# Variable Selection for Fault Detection Based on Causal Discovery Methods: Analysis of an Actual Industrial Case

Nayher Clavijo [1], Afrânio Melo [1], Rafael M. Soares [1], Luiz Felipe de O. Campos [1], Tiago Lemos [1], Maurício M. Câmara [1], Thiago K. Anzai [2], Fabio C. Diehl [2], Pedro H. Thompson [2] and José Carlos Pinto [1,*]

[1] Programa de Engenharia Química/COPPE, Universidade Federal do Rio de Janeiro, Cidade Universitária, CP 68502, Rio de Janeiro CEP 21941-972, RJ, Brazil; nayher@peq.coppe.ufrj.br (N.C.); afraeq@gmail.com (A.M.); rsoares@peq.coppe.ufrj.br (R.M.S.); luiz.felipe@coppe.ufrj.br (L.F.d.O.C.); tlemos@peq.coppe.ufrj.br (T.L.); mauricio@peq.coppe.ufrj.br (M.M.C.)

[2] Centro de Pesquisas Leopoldo Americo Miguez de Mello—CENPES, Petrobras—Petróleo Brasileiro SA, Rio de Janeiro CEP 21941-915, RJ, Brazil; tanzai@petrobras.com.br (T.K.A.); fabio.diehl@petrobras.com.br (F.C.D.); pedrothompson@petrobras.com.br (P.H.T.)

[*] Correspondence: pinto@peq.coppe.ufrj.br; Tel.: +55-21-3938-8337

**Abstract:** Variable selection constitutes an essential step to reduce dimensionality and improve performance of fault detection and diagnosis in large scale industrial processes. For this reason, in this paper, variable selection approaches based on causality are proposed and compared, in terms of model adjustment of available data and fault detection performance, with several other filter-based, wrapper-based, and embedded-based variable selection methods. These approaches are applied in a simulated benchmark case and an actual oil and gas industrial case considering four different learning models. The experimental results show that obtained models presented better performance during the fault detection stage when variable selection procedures based on causality were used for purpose of model building.

**Keywords:** fault detection and diagnosis; variable selection; feature selection; causality; conditional mutual information; real oil and gas process facility

## 1. Introduction

In the last decade, industrial process monitoring strategies have constantly evolved due to the technological improvements of sensors, equipment and instrumentation and, simultaneously, the increasing relevance of Industry 4.0 in the actual manufacturing process scenario [1]. As these complex industrial processes can produce large amounts of data, a large number of measured variables can be simultaneously monitored in modern plant-wide process monitoring platforms [2]. Hence, removing irrelevant and redundant variables constitutes an important data treatment step, simplifying data driven models, improving the process monitoring performance, and avoiding overfitting. In particular, Blum and Langley reviewed different definitions used to describe variable relevance in the machine learning literature [3].

Variable selection methodologies constitute an important approach to reduce dimensionality in fault detection and diagnosis problems and have become more relevant because of the recent significant increase of data-driven methods in this research area [4]. Variable selection algorithms normally try to identify the subset of measured variables that lead to the best analytical performance, being usually divided into three categories: filter-based, wrapper-based and embedded-based methods [3,5].

Filter-based methods do not depend on the employed learning algorithm and are often applied as a preprocessing step where the analyzed variables are ranked by relevance according to intrinsic properties of the data. This approach scores features in accordance with a certain statistical criterion, almost always making use of the $\chi^2$ statistics, $T$ statistics,

Pearson correlation, Spearman correlation, Fisher criteria, and metrics derived from the Information Theory. Some of these criteria have been revised by Ghosh et al. [6].

Wrapper-based methods explore the space of variable subsets and evaluate the performance of the models built with the subsets, consequently depending on the learning algorithm. These methods usually make use of one of the two following approaches: a sequential method, when selection starts with an empty set (full set), and add features (remove features) until satisfaction of a determined regression performance [7]; and the heuristic method, when the variable subsets are generated with help of a stochastic algorithm [8,9].

Embedded-based methods combine the learning model with an optimization problem, allowing variable selection and model building to be performed simultaneously. Differently from wrapper-based methods, embedded-methods incorporate the knowledge about a specific structure of the regression function in the variable selection engine. Two distinct families of embedded methods can be usually defined: the regularization methods [10,11] and the tree-based methods [12].

From the point of view of computational costs, filter-based methods are more attractive because of their inherent simplicity, as variable ranking can be established through simple score computations for each variable. Nevertheless, the variable subset found by these methods may not correspond to the subset that, jointly, maximize the classifier-regressor performance, since the variable relevance is affected neither by the model structure nor by the remaining process variables [13].

On the other hand, wrapper-based methods are more computationally intensive, but prevail over filter-based methods in terms of prediction accuracy since they take into account the classifier-regressor performance during the variable selection step [3,6]. One possible drawback of using wrapper-based methods is that the classifiers-regressors are prone to overfitting [14].

Finally, embedded-based methods try to compensate for the drawbacks discussed previously by incorporating the variable selection procedure as a part of the training process. However, application of these methods can be very intricate and limited to some specific learning models [5].

For all previously described strategies, an important aspect in the variable selection procedure is the criterion that defines the relevance of a single variable or subset of variables. Several criteria have been investigated and can be grouped into one or more of the following categories: distance, information, dependence, consistency and classifier error [15].

Mutual information (MI) is a measure of statistical independence that can be used to evaluate the relationship between random variables, including nonlinear relationships, being invariant under transformations of the feature space [16]. Distinct variable selection algorithms based on MI have been revised by Vergara and Estévez [17] using both filter-based methods [18,19] and wrapper-based methods [20,21]. In particular, Huang et al. [22] proposed a method where MI is used initially for variable ranking, while in a second step variable selection is guided by the maximization of information criterion. Relevant issues associated with the dimension of the selected variable subset and the mathematical connection between mutual information and regression performance metrics have been discussed in the literature [23,24]. Other metrics derived from MI, including the joint mutual information (JMI) [25], conditional mutual information (CMI) [26–28], and dynamic mutual information (DMI) [29], have also been studied. In particular, CMI, Granger metrics [30] and Transfer Entropy (TE) [31] can be used as causality measures and can be calculated from observed time series to characterize inner cause-effect relationships.

Based on the previous paragraphs, the present manuscript discusses and proposes the use of variable selection approaches based on time-lagged causality algorithms developed and applied for causal network reconstruction [32]. The relevance of variables are defined in accordance with their causal strength in respect to the predicted and monitored variable. Consequently, the proposed methodology allows isolating the partial effect of each variable in a set over the predicted variable, quantifying the amount of information shared condi-

tionally with the remaining variables. The causality quantification algorithms can adopt linear metrics (partial correlation) or nonlinear metrics (conditional mutual information). To validate the respective performance, results obtained with help of causality procedures are compared with results obtained with several other feature selection methods mentioned previously. Variable selection methodologies are applied in two scenarios:

1. a benchmark case, where the procedures are used to evaluate some simulated faults of the Tennessee-Eastman process.
2. a real industrial case, where the procedures are applied to actual industrial measurement datasets extracted from an oil and gas processing plant, with the objective to detect sensor faults reported by the operator.

The paper is structured as follows: in Section 2, we discuss information and causality theoretic preliminaries. In Section 3, the case studies and their respective faults scenarios are presented, and the research methodology is also discussed. In Section 4, we present and discuss several variable selection approaches applied in fault detection for both real (see Section 4.1) and artificial scenarios (see Section 4.2). This leads to recommendation of the use of variable selection methods based on causality approaches as discussed in Section 4.3. Finally, in Section 5, we conclude the paper and discuss future research.

## 2. Theoretical Background

### 2.1. Mutual Information and Entropy

The strategy for variable selection includes the identification of the input variables that contain the highest amount of information in relation to the output. Hence, entropy and mutual information are suitable measures in this context [23].

Entropy is a measure of uncertainty of a random variable. Considering $X_j$ as a discrete random variable, entropy can be defined as [33]:

$$H(X_j) = - \sum_{x_j \in X_j} p(x_j) log\, p(x_j) \tag{1}$$

where $x_j$ is the possible value of the random variable $X_j$, $p(x_j)$ is the probability density function of $x_j$.

For the case of two discrete random variables, i.e., $X_j$ and $Y_j$, the joint entropy of $X_j$ and $Y_j$ can be defined as follows [25]:

$$H(X_j, Y_j) = - \sum_{x_j \in X_j} \sum_{y_j \in Y_j} p(x_j, y_j) log\, p(x_j, y_j) \tag{2}$$

where $p(x_j, y_j)$ denotes the joint probability density function of $X_j$ and $Y_j$. Given that the value of another random variable $Y_j$ is known, the remaining uncertainty to describe the outcome of a random variable $X_j$ can be expressed by the conditional entropy [34]:

$$H(X_j|Y_j) = - \sum_{x_j \in X_j} \sum_{y_j \in Y_j} p(x_j, y_j) log\, p(x_j|y_j) \tag{3}$$

where $p(x_j|y_j)$ denotes the conditional probability density function of $X_j$ and $Y_j$. The amount of information that one variable provides about another one can be quantified by the mutual information (MI) [33]:

$$I(X_j, Y_j) = - \sum_{x_j \in X_j} \sum_{y_j \in Y_j} p(x_j, y_j) log\, \frac{p(x_j, y_j)}{p(x_j)} \tag{4}$$

Additionally, the MI and the entropy can be related as follows [33]:

$$I(X_j, Y_j) = H(X_j) - H(X_j, Y_j) \tag{5}$$

Mutual information can be interpreted as an independence or a correlation measure, being always non-negative, and equal to zero if and only if $X$ and $Y$ are independent [17].

*2.2. Conditional Mutual Information*

The conditional mutual information (CMI) can be given by [26]:

$$I(X_j, Y_j|Z_i) = \sum_{z_i \in Z_i} p(z_i) \sum_{x_j \in X_j} \sum_{y_j \in Y_j} p(x_j, y_j|z_i) log \frac{p(x_j, y_j|z_i)}{p(x_j|z_i)p(y_j|z_i)} \tag{6}$$

and measures the conditional dependence between $X_j$ and $Y_j$ given $Z_i$. The CMI can be interpreted as the MI between X and Y that is not contained in a third variable Z, and expressed in entropy terms as follows [26].

$$I(X_j, Y_j|Z_i) = H(X_j|Z_i) - H(X_j|, Y_j, Z_i) \tag{7}$$

*2.3. Conditional Independence and Causality*

An important task consists of quantifying the information flow in multiviariate systems. This quantification should be directed to meet the following tasks [35]: (1) quantification of linear-nonlinear associations and (2) characterization of the directionality of information flow propagation (causal interactions). These causal interactions can be visualized as links in an interaction network map.

According to Runge (2018) [36], a pair of variables (or nodes) $X_{t-\tau}^i$ and $X_t^j$ are connected by a direct causal link $X_{t-\tau}^i \rightarrow X_t^j$, for $\tau > 0$ if and only if

$$X_{t-\tau}^i \not\perp X_t^j | \mathbf{X}_t^- \setminus X_{t-\tau}^i \tag{8}$$

so that they are not independent conditionally over the past of the whole multivariable system (process) $\mathbf{X}_t^-$ excluding $X_{t-\tau}$. Here it is assumed that the multivariable system $\mathbf{X}$ contains $N$ variables $= (X^{i=1}, X^{i=2}, ..., X^{i=N}, ...)$. The past of the entire system is denoted as $\mathbf{X}_t^- = (\mathbf{X}_{t-1}, \mathbf{X}_{t-2}, ..., \mathbf{X}_{t-\tau_{max}})$, where the subset $\mathbf{X}_{t-\tau}$ is composed by the lagged variables $(X_{t-\tau}^{i=1}, X_{t-\tau}^{i=2}, ..., X_{t-\tau}^{i=N})$. When $X_t^i = X_t^j$, this measure represents an autodependency at lag $\tau$. Moreover, the set of parents of a variable (node) $X_t^j$ can be defined by [36]:

$$\mathcal{P}_{X_t^j} \equiv \left\{ Z_{t-\tau} : Z \in \mathbf{X}, \ \tau > 0, \ Z_{t-\tau} \rightarrow X_t^j \right\} \tag{9}$$

The parents of all variables (subprocesses) in $\mathbf{X}$ and the contemporaneous links comprise the time series graph [32].

Characterization of causal links (Equation (8)) can be performed with different linear or nonlinear independence measures. In particular, MI constitutes an important metric to measure linear and nonlinear associations between variables, but not the direction of dependence. The Granger causality [30] and, in a more general context, the Transfer Entropy (TE) [31] can provide practical means to satisfy these tasks [36].

$$I_{X^i \rightarrow X^j}^{TE} = I \left( X_{t-\tau}^i; X_t^j | \mathbf{X}_t^- \setminus X_{t-\tau}^i \right) \tag{10}$$

Here TE is expressed in CMI terms and measures the aggregated influence of $X^i$ over all past lags, but lead to a problem when high-dimensional probability density functions (PDF) must be estimated [37]. Lag-specific variants of TE (relative information transfer, and momentary information transfer) have been introduced [32,35] to avoid the computation of PDFs of high dimensions. An analogous form of Equation (10) can be obtained by substituting CMI (nonlinear independence measure) by the linear partial correlation (linear independence measure) term [32].

### 2.4. Approaches

In this present work, causal links characterization algorithms are used in the variable selection context. Some of these algorithms are shortly described in the following sections.

#### 2.4.1. PC-Stable Algorithm

The PC algorithm [38] is a well-known causal link characterization algorithm, widely used to reconstruct causal relationships which can be represented by a Directed Acyclic Graph (DAG). The algorithm consists of an iterative procedure where pairs of variables (at different time lags) conditionally independent (at some significance level) are estimated. The lagged links, computed according to Equation (10), provides the strength and orientation of these causal links. In the present work, a robust modification of the PC algorithm called PC-stable [39] is used.

In particular, the PC algorithm evaluates and removes links from the DAG and updates the network dynamically. Therefore, the resulting network is dependent on the order in which the conditional independence tests are performed. On the other hand, the PC Stable algorithm prevents the link deletion affecting the conditioning set Z of the other variables. A schematic example [40] of the PC algorithm and the PC Stable algorithm applications are presented in Appendix A.1 to introduce the main aspects and differences of the algorithms.

Algorithm 1 summarises the procedures of the PC-stable method as applied in the present work. A detailed description of this algorithm can be found in the original references [39,41,42].

---

**Algorithm 1:** PC-stable algorithm

---

**Input:** Dataset with set of variables $\mathbf{X}$, maximum time lag $\tau_{max}$ and a significant level $\alpha$
**Output:** List of parents of desired-output variable $X_t^j$
**Result:** Parents sorted by dependence, $\mathcal{P}_{X_t^j}$

Assume all variables (nodes) are connected initially;
**for** $X_{t-\tau}^i$ *in* $\mathbf{X}$ **do**
  Compute all pairwise dependence $X_{t-\tau}^i \not\perp X_t^j$;
  **if** $X_{t-\tau}^i \not\perp X_t^j$ *is significant, i.e pvalue* $< \alpha$ **then**
    add $X_{t-\tau}^i$ to initial parent list $\mathcal{P}_{X_t^j}$
  **end**
**end**
given the initial list of parents $\mathcal{P}_{X_t^j} = \left[ X_{t-\tau}^i : i = 1,..,N, \tau = 1,...,\tau_{max} \right]$;
Let max conditions dimension $d_{max} = 0$;
**while** $d_{max} < length(\mathcal{P}_{X_t^j})$ **do**
  **for** $X_{t-\tau}^i$ *in* $\mathcal{P}_{X_t^j}$ **do**
    **while** *still untested conditions* $\mathcal{S} \subseteq \mathcal{P}_{X_t^j}$ **do**
      Select a new condition $\mathcal{S}$ from the sorted list of parents $\mathcal{P}_{X_t^j}$ ;
      Considering the link $X_{t-\tau}^i \to X_t^j$ Test $X_{t-\tau}^i \not\perp X_t^j | \mathcal{S}$;
      **if** $X_{t-\tau}^i \to X_t^j$ *is not a significant link, i.e pvalue* $> \alpha$ **then**
        Break inner while loop;
      **end**
    **end**
  **end**
  Let $d = d + 1$ Remove non-significant parents from $\mathcal{P}_{Y_t}$;
  Sort parents (descending order) according value of conditional dependence metric.
**end**

---

### 2.4.2. PCMCI Algorithm

Another causal link characterization algorithm is the PCMCI algorithm [36], which is aimed to circumvent some PC-stable limitations related to the optimal selection of conditioning sets, improving the accuracy of independence strength estimation. Briefly, the PCMCI algorithm considers two stages:

1.  Estimate the parents $\mathcal{P}_{X_t^j}$ for every variable $X_t^j \in \mathbf{X}$ using the PC-Stable algorithm.
2.  Using the estimated set of parents, perform a novel independence test called momentary conditional independence (MCI), where given the variable pair $(X_{t-\tau}^i, X_t^j)$:

$$MCI : X_{t-\tau}^i \not\perp\!\!\!\perp X_t^j | \mathcal{P}_{X_t^j} \setminus \left\{ X_{t-\tau}^i \right\}, \mathcal{P}_{X_{t-\tau}^i} \tag{11}$$

Theoretical description and practical applications of the PCMCI algorithm have been thoroughly discussed elsewhere [36,41].

## 3. Case Studies

### 3.1. Benchmark Case: Tennessee-Eastman Process

The Tennessee-Eastman Process (TEP) [43] is a widely used benchmark model, which serves as the industrial basis to assess the capability of fault detection and diagnosis methods. In the TEP process, there are five major units: reactor, condenser, compressor, separator and stripper. The process provides two products from four reactants. Also, an inert and a by-product are present in the process streams, leading to a total of 8 components denoted as A, B, C, D, E, F, G and H. TEP covers 22 process variables and 12 manipulated variables, resulting in 34 measured variables. Typically, measurements related to the mole fraction of components in the reactor feed, purge gas and product streams are not considered because their characteristic sampling intervals are too long [44]. A schematic diagram of the TEP process and the complete list of variables are presented in Figure A3 and Appendix A.2.

The benchmark presents 20 faults, which were originally defined by Downs and Vogel [43], and an additional valve fault further introduced in Chiang et al. [44]. The dataset used in the present work was generated with the original FORTRAN code available at http://brahms.scs.uiuc.edu (accessed 13 March 2018).

In the present work, 2 out of the 21 available faults were considered to validate the proposed variable selection methods. Table 1 summarizes the analyzed TEP faults.

**Table 1.** Analyzed TEP process faults (see Appendix A.2).

| Fault Number | Process Variable | Type | Monitored Variable |
|:---:|:---:|:---:|:---:|
| IDV(1) | A/C feed ratio, B composition constant | Step | XMEAS(23) |
| IDV(5) | Condenser cooling water inlet temperature | Step | XMEAS(22) |

### 3.2. Real Industrial Case: Oil and Gas Fiscal Metering Station

The industrial data used in the present work were acquired with helpf of online sensors of an onshore metering station located in a Petrobras field.

Briefly, the industrial process is composed of three fiscal metering stations (two gas fiscal metering stations and an oil fiscal metering station). Figure 1 describes shortly the oil and gas fiscal metering process and enumerates its respective sections. A more detailed description of this process, the observed faults and their respective phenomenological and economic consequences during the custody transfer process have been discussed elsewhere [45].

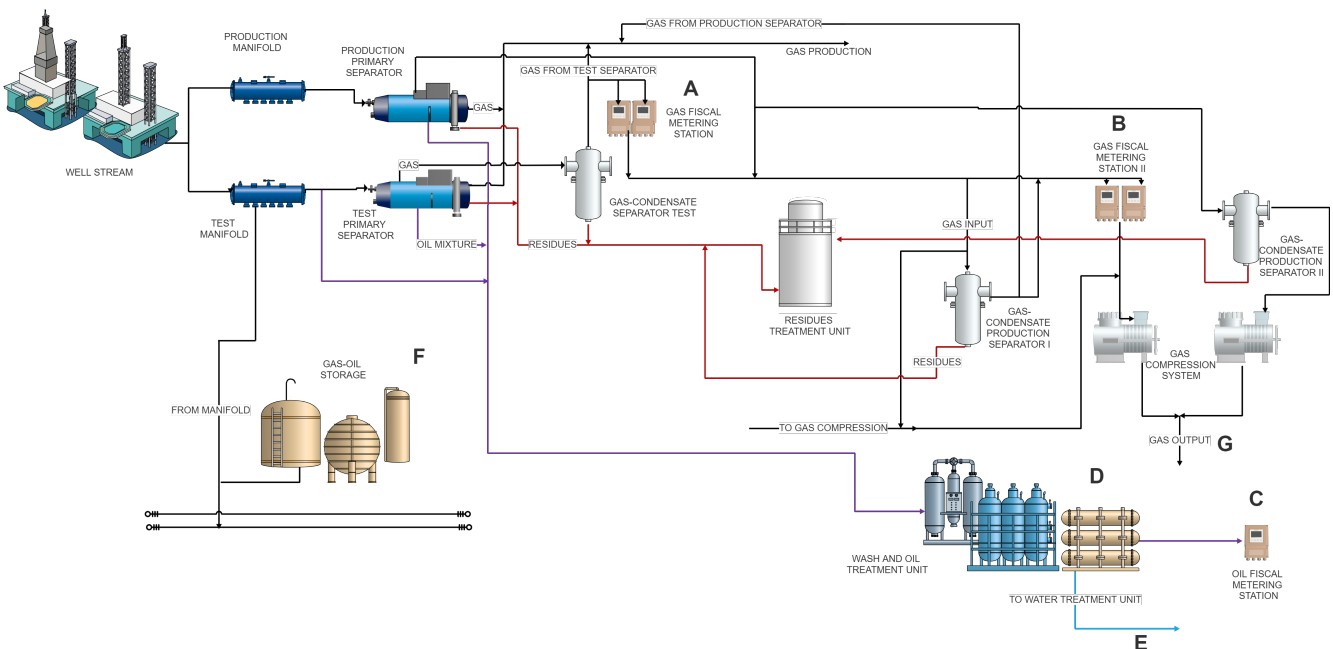

**Figure 1.** Oil and gas metering station PFD.

The dataset used in the present work comprises measurements of 112 process variables, collected at frequency $f = 1min$ during three consecutive years. Table 2 summarizes these process variables.

**Table 2.** Variable measurements in the gas-oil metering station (see Figure 1).

| Variable Type | Number of Measurements |
|---|---|
| Flow rate | 40 |
| Temperature | 11 |
| Controller output | 2 |
| Differential pressures | 8 |
| Pressures | 21 |
| Levels | 2 |
| Relative density | 9 |
| BSW (water content) | 6 |
| Electric current | 9 |
| Valve aperture | 4 |

Several types of faults were identified in the fiscal sensors of the process plant. Most of them were related to faults of temperature sensors and flow sensors. Table 3 shows the faults studied in the present work and the respective dataset sizes for training, validation, and testing of the employed regressor models. In particular, detections of faults F-II and F-III were performed with the same training data.

**Table 3.** Faults description.

| Fault | Training Set Size (Points) | Validation Set Size (Points) | Test Set Size (Points) | Monitored Variable |
|---|---|---|---|---|
| F-I | 20,161 | 120,056 | 42,660 | Gas flow rate in B |
| F-II | 44,581 | 106,620 | 41,760 | Gas temperature in A |
| F-III | 44,581 | 74,727 | 2880 | Gas temperature in A |

*3.3. Methodology*

The fault detection approach used in the present work is discrepancy-based, where the fault pattern is recognized using a residual measure calculated as the difference between the process sensors readings and the expected values obtained using the predictions of a model that represents the process in fault-free conditions. In the present work, the model building is based on unsupervised learning techniques (regression models) since the number of failures in each sensor and the constant process evolution makes the use of supervised techniques (classification models) unfeasible. The purpose of the unsupervised techniques is to model, based on the input and output datasets ($X$ and $Y$, respectively), the dynamic behavior of the process in normal (non-fault) conditions. The process condition monitoring is based on a residual metric calculation. In the present case, the Square Prediction Error (SPE) was used, which is a fault detection index that allows the identification of abnormal conditions along the control chart.

Here, the fault detection models were configured considering a single output $Y$ as the fault observed variable (See monitored variable in Tables 1 and 3). On the other hand, the input variable (set $X$) was generated applying the variable selection procedures (listed in Table 4) on the original complete dataset. Moreover, the number of variables in the input set $X$ was previously established as the number of principal components needed to describe 95% of the cumulative variance in the respective training set.

**Table 4.** Applied variable selection methods.

| Variable Selection Method | Class of Method |
|:---:|:---:|
| Pearson correlation-based | Filter |
| Spearman correlation-based | Filter |
| Mutual information-based | Filter |
| Forward feature elimination (Lasso) | Wrapper |
| Forward feature elimination (Random Forest) | Wrapper |
| Backward feature elimination (Lasso) | Wrapper |
| Backward feature elimination (Random Forest) | Wrapper |
| L1-Regularization Lasso-based | Embedded |
| Random Forest importance-based | Embedded |
| PCMCI (partial correlation) | Filter |
| PCStable (partial correlation) | Filter |
| PCStable (partial correlation) + MCI (conditional mutual information) | Filter |

To evaluate the fault detection performance in the above-reported case studies, the machine learning regressors described in Table 5 were considered, which were evaluated for each input subset $X$ generated by the variable selection methods. The architecture and respective hyperparameters were defined according to heuristics reported in the literature, also summarized in Table 5. Thus, the efficiencies of the variable selection algorithms were directly associated with the fault detection performance of the models trained with the same hyperparameter values, allowing observation of the variable selection effect with small influence of the hyperparameter values.

**Table 5.** Heuristics for setting hyperparameter values during unsupervised model regressions.

| Regressor | Hyperameter Heuristics |
|---|---|
| Canonical correlation analysis (CCA) | *Number of components*: the number of principal components needed to describe 95% of cumulative correlation. |
| Ridge regression (RR) | *Regularization strength* ($\alpha$): A cross validation procedure was required to determine the value of this parameter. |
| Multilayer perceptron regressor (MLPR) | *Number of hidden layers*: Regression problems that require two or more hidden layers are rarely encountered. Network architectures with one hidden layer can approximate any function that contains a continuous mapping from one finite space to another [46]. *Number of neurons in the hidden layers*: Specifying as many hidden nodes as dimensions (principal components) needed to capture 70–90% of the variance of the input dataset [47–49]. *Activation function*: ReLU is a general activation function widely used in regression problems [50,51]. |
| Random forest regressor (RF) | *Number of estimators*: Sometimes the use of a large number of trees in the random forest does not lead to any significant performance gain. Previous benchmarks evaluation suggests a range between 64 and 128 trees in a forest [52,53]. *Number of features to consider when looking for the best split*: On average, empirical results have shown good results with sqrt(Number of features) and (Number of features) for classification and regression problems, respectively [54]. *Bootstrap*: To improve the robustness of forecast, use of bootstrap sampling is recommended when building trees [55]. |

Overall, we considered 12 selection variable methods corresponding of which there were three filter-based, four wrapped-based, two embedded-based, and three causality-based ones. Each selection variable method was applied in five fault detection scenarios, three faults cases of the actual industrial case and two faults cases of TEP, using four different machine learning models. Furthermore, references fault detection scenarios for the actual industrial process and TEP were obtained considering the original complete dataset (i.e., without taking into account procedures of variable selection).

## 4. Results

In this section, the effectiveness of the proposed causal link characterization approaches are evaluated as variable selection methods. Other variable selection algorithms were considered in order to compare and discuss effects on fault detection cases. Results and discussions regarding real industrial and benchmark datasets are shown in Sections 4.1 and 4.2, respectively.

### 4.1. Performance on Real Industrial Case

In the present work, the subsets of selected variables we assumed to have a fixed size. An estimate of the number of variables to be selected by the variable selection procedures can be calculated through principal component analysis (PCA). Thus, considering

a cumulative explained variance of 95.0%, the number of required principal components corresponded to a total of 20 variables. The complete analysis is shown in Figure A4 present in Appendix A.3.

The performance of the analyzed variable selection approaches are characterized in terms of the following performance metrics: fault detection rate (FDR %), false alarm rate (FAR %) and regression score $R^2$. To establish a reference point for all the studied faults, the learning models were also trained without taking into account procedures of variable/feature selection. Table 6 shows the respective results that will be considered as the reference performance values for comparison with the performance of the models trained with use of variable selection methods. The regressor predictions obtained with these models for Faults I, II, and III are presented in Figures A6–A8 in Appendix A.4.

**Table 6.** Reference performance when variable selection procedures were not used to analyze the industrial data.

| Fault | Regressor | FDR (%) | FAR (%) | $R^2$ Training Set | $R^2$ Validation Set | $R^2$ Test Set |
|-------|-----------|---------|---------|-----------|-----------|-----------|
| F-I | RR | 0.0 | 10.71 | 0.99 | −186.93 | −690.37 |
| | RF | 8.4 | 10.42 | 0.99 | 0.96 | −24.26 |
| | MLPR | 0.0 | 10.59 | 0.95 | −23.69 | −78.41 |
| | CCA | 0.0 | 10.60 | 0.99 | −170.23 | −635.27 |
| F-II | RR | 3.98 | 0.0 | 0.88 | −21.76 | −1.78 |
| | RF | 59.4 | 0.0 | 1.0 | −0.34 | −0.95 |
| | MLPR | 10.96 | 0.0 | 0.76 | −19.07 | −2.31 |
| | CCA | 21.53 | 0.0 | 0.43 | −2.79 | −0.40 |
| F-III | RR | 51.47 | 11.0 | 0.88 | −20.28 | −0.61 |
| | RF | 63.04 | 7.29 | 1.0 | −0.14 | −0.18 |
| | MLPR | 8.87 | 0.0 | −1.41 | −185.67 | −1.09 |
| | CCA | 63.44 | 0.3 | 0.43 | −0.51 | −0.22 |

The performance metrics corresponds to fault detection rate (FDR%), false alarm rate (FAR%) and regression score $R^2$.

Table 7 shows the performance of the regressors when filter-based variable selection methods were used. In general, the regressors were able to detect Fault F-III, but unable to detect Fault F-I. On the other hand, Fault F-II led to the highest detection rates (FDR %) when the variable selection method was based on mutual information. As one can see, the learning models that used variable selection procedures based on linear correlation (Pearson and Spearman) were more likely to present overfitting, as $R^2$ values for the validation set were negative. However, lower values of $R^2$ for Fault F-I validation set were expected because this set was much larger than the test set and, chronologically, was the most distant from the fault event, incorporating dynamic behaviors that had not been possibly captured in the training set. As it might already be expected, $R^2$ values were obtained in the test because of the presence of many faulty data.

**Table 7.** Performance when filter-based variable selection procedures were used to analyze the industrial data.

| Variable Selection Method | Fault | Regressor | FDR (%) | FAR (%) | $R^2$ Training Set | $R^2$ Training Set | $R^2$ Training Set |
|---|---|---|---|---|---|---|---|
| Pearson-based | F-I | RR | 0.0 | 0.02 | 0.85 | −153700 | −379953 |
| | | RF | 9.1 | 0.03 | 0.99 | 0.71 | 0.709 |
| | | MLPR | 0.0 | 0.0 | 0.64 | −377 | −928.18 |
| | | CCA | 0.0 | 0.03 | 0.54 | −63786 | −63786 |
| | F-II | RR | 1.9 | 8.12 | 0.79 | −41.74 | −11.26 |
| | | RF | 74.5 | 0.0 | 0.99 | 0.04 | −0.69 |
| | | MLPR | 0.0 | 0.0 | 0.19 | −121.6 | −1.32 |
| | | CCA | 1.9 | 8.24 | 0.79 | −42.74 | −11.67 |
| | F-III | RR | 63.4 | 0.0 | 0.79 | −22.59 | −0.30 |
| | | RF | 63.4 | 23.14 | 0.99 | −0.35 | −0.23 |
| | | MLPR | 63.4 | 0.0 | 0.68 | 0.32 | 0.32 |
| | | CCA | 63.4 | 0.0 | 0.79 | −22.84 | −0.30 |
| Spearman based | F-I | RR | 0.0 | 0.01 | 0.83 | −85700 | −241433 |
| | | RF | 6.1 | 0.05 | 0.98 | 0.78 | 0.512 |
| | | MLPR | 9.8 | 0.02 | 0.96 | −2.31 | −6.061 |
| | | CCA | 0.0 | 0.03 | 0.54 | −26281 | −63786 |
| | F-II | RR | 2.2 | 6.95 | 0.72 | −40.41 | −11.02 |
| | | RF | 78.5 | 0.0 | 0.99 | 0.08 | −0.47 |
| | | MLPR | 4.9 | 0.0 | 0.78 | −35.79 | −0.64 |
| | | CCA | 1.97 | 8.24 | 0.79 | −42.74 | −11.72 |
| | F-III | RR | 60.6 | 0 | 0.68 | −18.42 | −0.12 |
| | | RF | 63.4 | 16.47 | 0.98 | −0.19 | −0.21 |
| | | MLPR | 63.4 | 0.16 | 0.57 | −214.65 | −2.74 |
| | | CCA | 63.4 | 0.0 | 0.79 | −22.84 | −0.31 |
| Mutual information-based | F-I | RR | 11.0 | 10.51 | 0.99 | 0.69 | −411.87 |
| | | RF | 28.4 | 10.43 | 0.99 | 0.99 | −32.64 |
| | | MLPR | 12.1 | 10.42 | 0.90 | −2.66 | −38.65 |
| | | CCA | 9.1 | 10.61 | 0.97 | 0.81 | −350.76 |
| | F-II | RR | 6.9 | 0.0 | 0.56 | −244.78 | −8.21 |
| | | RF | 78.4 | 3.05 | 0.99 | −0.04 | −0.83 |
| | | MLPR | 10.7 | 0.0 | 0.43 | −11.15 | −0.15 |
| | | CCA | 26.8 | 0.0 | 0.20 | −14.72 | −1.26 |
| | F-III | RR | 63.4 | 0.0 | 0.56 | −64.56 | −1.47 |
| | | RF | 63.4 | 0.0 | 0.99 | −0.25 | −0.21 |
| | | MLPR | 63.4 | 0.0 | 0.75 | −26.28 | −0.41 |
| | | CCA | 63.4 | 0.0 | 0.20 | −2.12 | −0.32 |

The performance metrics corresponds to fault detection rate (FDR%), false alarm rate (FAR%) and regression score $R^2$.

Considering the average performance of the 4 regressors, the highest FDR values and lowest FAR values were achieved when the mutual information-based variable selection method was used. This can possibly be explained because the mutual information metric is

able to capture nonlinear associations among the variables, while the Pearson or Spearman correlations are unable to detect these nonlinear associations.

When compared to the reference performance, the methods based on linear correlations (Pearson and Spearman) led to worse results in the three faults, while the method based on mutual information was better in the three cases.

The regressor performance obtained when wrapper-based variable selection procedures were used are summarized in Table 8. It is possible to observe that Fault F-III was properly detected with all analyzed wrapper-based variable selection methods. On the other hand, Fault F-I was not detected, except when the Random Forest model was used, while the best detections of the F-II fault were achieved with the variable selection procedure based on the forward feature elimination (Lasso) followed by the backward feature elimination (Lasso). As it might be expected, high FDR (%) and $R^2$ values were obtained with the training and validation sets when the learning model in the wrapper method coincided with the regressor model (Random Forest). Another aspect that must be highlighted regards the general performance of wrapper methods, which achieved higher $R^2$ values than filter methods. Regressors trained with use of wrapper methods presented better ability to correctly model new data (generalization), as observed in the regression scores of the validation sets. In addition, only the wrapper methods that used Lasso learning model exceeded the reference performance in all faults detection scenarios.

Table 9 presents the regressor performance obtained with embedded-based variable selection procedures. On the whole, although Fault F-III was always properly identified, these regressors showed lower rates of failure detection than described previously for wrapper-based variable selection approaches. Besides, the selection procedures based on random forest schemes provided poorer models that were subject to overfitting. In general, the learning models that considered a variable selection step based on embedded methods did not show substantial improvements when compared to the reference performances.

**Table 8.** Performance when wrapper-based variable selection procedures were used to analyze the industrial data.

| Variable Selection Method | Fault | Regressor | FDR (%) | FAR (%) | $R^2$ Training Set | $R^2$ Training Set | $R^2$ Training Set |
|---|---|---|---|---|---|---|---|
| Forward feature elimination (Lasso) | F-I | RR | 0.4 | 11.43 | 0.98 | −15.90 | −1506.12 |
| | | RF | 8.6 | 10.42 | 0.99 | 0.99 | −28.19 |
| | | MLPR | 0.0 | 10.52 | 0.99 | −25.00 | −268.13 |
| | | CCA | 0.3 | 11.45 | 0.98 | −15.49 | −1535 |
| | F-II | RR | 5.4 | 11.93 | 0.80 | −23.24 | −7.70 |
| | | RF | 57.1 | 0.0 | 0.99 | −0.33 | −0.56 |
| | | MLPR | 5.2 | 10.28 | 0.65 | −13.90 | −2.34 |
| | | CCA | 5.3 | 11.53 | 0.70 | −10.22 | −2.18 |
| | F-III | RR | 63.4 | 0.0 | 0.80 | −3.71 | −0.19 |
| | | RF | 63.4 | 6.62 | 0.99 | −0.25 | −0.21 |
| | | MLPR | 63.4 | 0.0 | 0.47 | −8.21 | 0.05 |
| | | CCA | 63.4 | 0.0 | 0.70 | −1.21 | −0.16 |

**Table 8.** *Cont.*

| Variable Selection Method | Fault | Regressor | FDR (%) | FAR (%) | $R^2$ Training Set | $R^2$ Training Set | $R^2$ Training Set |
|---|---|---|---|---|---|---|---|
| Forward feature elimination (Random Forest) | F-I | RR | 8.0 | 10.50 | 0.98 | −29.41 | −876.13 |
| | | RF | 23.7 | 10.43 | 0.99 | 0.97 | −38.19 |
| | | MLPR | 7.4 | 0.78 | 0.89 | −46.03 | −59.49 |
| | | CCA | 10.9 | 10.5 | 0.94 | −46.71 | −1268 |
| | F-II | RR | 19.68 | 4.58 | 0.71 | −2.17 | −1.85 |
| | | RF | 68.57 | 0.0 | 0.99 | −1.03 | −1.31 |
| | | MLPR | 11.15 | 3.31 | 0.47 | −4.31 | −0.16 |
| | | CCA | 31.60 | 0.31 | 0.57 | −0.12 | −1.08 |
| | F-III | RR | 63.4 | 0.0 | 0.71 | −128.02 | −2.01 |
| | | RF | 63.4 | 0.0 | 0.99 | −1.25 | −0.22 |
| | | MLPR | 63.4 | 0.0 | 0.52 | −182.21 | −0.48 |
| | | CCA | 63.4 | 0.0 | 0.54 | −35.93 | −0.85 |
| Backward feature elimination (Lasso) | F-I | RR | 0.0 | 0.0 | 0.31 | −424 | −247.82 |
| | | RF | 0.3 | 2.41 | 0.99 | −6.88 | −38.04 |
| | | MLPR | 0.0 | 0.06 | −0.01 | −9.57 | −23.18 |
| | | CCA | 0.0 | 0.0 | 0.22 | −1106 | −611.13 |
| | F-II | RR | 21.5 | 24.51 | 0.75 | −20.72 | −4.21 |
| | | RF | 60.5 | 0.0 | 0.99 | −1.84 | −0.73 |
| | | MLPR | 11.4 | 0.0 | −1.05 | −23.98 | −0.43 |
| | | CCA | 21.4 | 25.51 | 0.70 | −34.81 | −7.81 |
| | F-III | RR | 63.4 | 1.60 | 0.75 | −9.94 | −0.76 |
| | | RF | 63.4 | 0.0 | 0.99 | −1.28 | −0.23 |
| | | MLPR | 63.4 | 0.0 | 0.01 | −28.34 | −0.75 |
| | | CCA | 63.4 | 1.60 | 0.70 | −17.06 | −1.01 |
| Backward feature elimination (Random Forest) | F-I | RR | 0.0 | 0.14 | 0.91 | −27.23 | −25.16 |
| | | RF | 10.4 | 0.08 | 0.99 | 0.78 | 0.66 |
| | | MLPR | 0.0 | 0.03 | 0.76 | −20.81 | −66.90 |
| | | CCA | 0.0 | 0.15 | 0.89 | −35.51 | −30.75 |
| | F-II | RR | 4.9 | 14.47 | 0.79 | −24.14 | −8.20 |
| | | RF | 68.2 | 0.0 | 0.99 | −0.16 | −0.67 |
| | | MLPR | 1.5 | 0.0 | 0.19 | −22.97 | −0.05 |
| | | CCA | 5.8 | 14.38 | 0.78 | −28.84 | −10.25 |
| | F−III | RR | 63.4 | 0.0 | 0.79 | −7.31 | −0.33 |
| | | RF | 63.4 | 18.34 | 0.99 | −0.48 | −0.18 |
| | | MLPR | 63.4 | 53.36 | 0.35 | −24.15 | −1.51 |
| | | CCA | 63.4 | 0.0 | 0.78 | −8.67 | −0.36 |

The performance metrics corresponds to fault detection rate (FDR%), false alarm rate (FAR%) and regression score $R^2$.

**Table 9.** Performance when embedded-based variable selection procedures were used to analyze the industrial data.

| Variable Selection Method | Fault | Regressor | FDR (%) | FAR (%) | $R^2$ Training Set | $R^2$ Training Set | $R^2$ Training Set |
|---|---|---|---|---|---|---|---|
| L1-regularization (Lasso) | F-I | RR | 0.0 | 0.02 | 0.91 | −94.52 | −189.52 |
| | | RF | 0.7 | 0.07 | 0.99 | 0.76 | 0.71 |
| | | MLPR | 0.0 | 1.55 | 0.88 | −4.75 | −19.70 |
| | | CCA | 0.0 | 0.02 | 0.84 | −51.24 | −96.02 |
| | F-II | RR | 7.6 | 12.28 | 0.81 | −51.27 | −19.62 |
| | | RF | 58.9 | 0.0 | 0.99 | −0.26 | −0.51 |
| | | MLPR | 44.35 | 4.71 | 0.79 | −5.64 | −2.46 |
| | | CCA | 7.8 | 12.30 | 0.81 | −52.98 | −52.98 |
| | F-III | RR | 63.7 | 1.36 | 0.81 | −9.39 | −0.37 |
| | | RF | 63.4 | 0.0 | 0.99 | −0.71 | −0.21 |
| | | MLPR | 63.4 | 0.0 | 0.04 | −21.32 | −0.41 |
| | | CCA | 63.7 | 1.28 | 0.81 | −9.76 | −0.38 |
| Random forest importances | F-I | RR | 0.0 | 0.12 | 1.0 | 1.0 | 1.0 |
| | | RF | 0.3 | 0.08 | 0.99 | 0.99 | 0.94 |
| | | MLPR | 0.0 | 11.60 | 0.89 | −2.65 | −15.78 |
| | | CCA | 81.1 | 0.20 | 1.0 | 1.0 | 1.0 |
| | F-II | RR | 0.6 | 0.0 | 1.0 | 0.99 | 0.99 |
| | | RF | 69.2 | 0.0 | 0.99 | 0.99 | 0.12 |
| | | MLPR | 0.0 | 0.0 | −0.97 | −1257 | −3.59 |
| | | CCA | 26.1 | 17.58 | 1.0 | 1.0 | 1.0 |
| | F-III | RR | 63.4 | 0.0 | 1.0 | 0.99 | 0.99 |
| | | RF | 63.4 | 2.23 | 0.99 | 0.99 | 0.01 |
| | | MLPR | 0.0 | 0.0 | 0.82 | −335.12 | −3.09 |
| | | CCA | 63.4 | 0.80 | 1.0 | 1.0 | 1.0 |

The performance metrics corresponds to fault detection rate (FDR%), false alarm rate (FAR%) and regression score $R^2$.

Although variable selection methods based on causal relationships were classified as filter methods, Table 10 shows the independent evaluation of the respective fault detection results obtained with these methods. As one can see, causality-based approaches outperformed the other methods when tested with most of the faults in terms of selecting the subset that produces the best regression accuracy. These approaches also led to the best $R^2$ values for the validation set, generating more generalistic learning models and providing on average the highest FDR and lowest FAR values among all methods applied here. This better generalization capability proved to be fundamental in the analyzed context because the process is likely to be subject to dynamic changes during the operation time as a function of the variations on the plant operating conditions. In particular, the PCMCI procedure, with PCStable stage using partial correlation and MCI stage using conditional mutual information metrics, proved to be the most suitable procedure for the detection of Faults II and III, while the best Fault I detection performance was achieved using the PCMCI procedure considering partial correlation metrics in its two stages.

**Table 10.** Performance when causality methods were used for variable selection analysis of the industrial data.

| Variable Selection Method | Fault | Regressor | FDR (%) | FAR (%) | $R^2$ Training Set | $R^2$ Training Set | $R^2$ Training Set |
|---|---|---|---|---|---|---|---|
| PCMCI (Partial corellation) | F-I | RR | 11.1 | 10.36 | 0.98 | 0.21 | −218.91 |
| | | RF | 15.4 | 10.76 | 0.99 | 0.85 | −33.96 |
| | | MLPR | 38.1 | 21.02 | 0.91 | −2.31 | −237.33 |
| | | CCA | 66.4 | 10.41 | 0.86 | 0.61 | −92.75 |
| | F-II | RR | 0.0 | 0.0 | 0.74 | −439.89 | −3.41 |
| | | RF | 79.7 | 0.0 | 0.99 | −0.82 | −0.97 |
| | | MLPR | 0.7 | 18.04 | 0.05 | −105.12 | −7.64 |
| | | CCA | 44.7 | 0.0 | 0.01 | −0.50 | −0.44 |
| | F-III | RR | 42.2 | 0.08 | 0.75 | −312.6 | −4.03 |
| | | RF | 63.4 | 0.0 | 0.99 | −0.36 | −0.21 |
| | | MLPR | 4.3 | 2.04 | 0.40 | −1311 | −20.17 |
| | | CCA | 63.4 | 0.0 | 0.02 | −0.57 | −0.11 |
| PCStable (Partial correlation) | F-I | RR | 1.1 | 10.61 | 0.98 | 0.60 | −187.87 |
| | | RF | 3.8 | 10.85 | 0.99 | 0.90 | −27.37 |
| | | MLPR | 0.4 | 10.45 | 0.95 | −0.02 | −81.35 |
| | | CCA | 10.5 | 10.51 | 0.85 | 0.47 | −84.97 |
| | F-II | RR | 8.67 | 0.0 | 0.55 | −15.23 | −0.61 |
| | | RF | 74.9 | 0.0 | 0.99 | −0.40 | −0.75 |
| | | MLPR | 0.0 | 0.0 | 0.40 | −501.12 | −4.95 |
| | | CCA | 34.1 | 0.0 | 0.01 | −0.89 | −0.44 |
| | F-III | RR | 63.4 | 0.08 | 0.55 | −45.23 | −0.60 |
| | | RF | 63.3 | 10.36 | 0.99 | −1.23 | −0.25 |
| | | MLPR | 63.4 | 0.0 | 0.37 | −9.96 | −0.67 |
| | | CCA | 63.9 | 1.12 | 0.01 | −0.85 | −0.11 |
| PCStable (Partial correlation) + MCI (Conditional mutual information) | F-I | RR | 10.1 | 10.56 | 0.98 | 0.68 | −275.59 |
| | | RF | 10.38 | 10.86 | 0.99 | 0.90 | −26.26 |
| | | MLPR | 10.4 | 10.47 | 0.97 | 0.80 | −498.66 |
| | | CCA | 13.9 | 10.51 | 0.92 | 0.60 | −158.31 |
| | F-II | RR | 28.8 | 0.0 | 0.57 | −0.55 | −0.31 |
| | | RF | 61.3 | 0.0 | 0.99 | −0.06 | −0.07 |
| | | MLPR | 21.7 | 0.1 | 0.62 | −0.93 | −0.28 |
| | | CCA | 49.3 | 0.0 | 0.42 | −0.02 | −0.35 |
| | F-III | RR | 63.4 | 0.0 | 0.57 | −2.97 | −0.17 |
| | | RF | 63.7 | 9.78 | 0.99 | −0.24 | −0.18 |
| | | MLPR | 63.4 | 0.0 | 0.37 | −5.62 | −0.18 |
| | | CCA | 63.4 | 0.0 | 0.45 | −1.13 | −1.14 |

The performance metrics corresponds to fault detection rate (FDR%), false alarm rate (FAR%) and regression score $R^2$.

Figure 2 shows the predictions of Fault F-I obtained with PCMCI (partial correlation). For all analyzed regressors, it is possible to observe good $R^2$ values for the training and validation sets and a clear divergence between measured data and respective predictions

in the test set near the failure event. Figure 3 presents the respective SPE index plot, where regression residues in the training and validation sets remained below the control limit, except for some sporadic points which were responsible for the observed FAR rates. This control limit was exceeded consistently in the reported fault event, proving the capacity of these models for fault detection. As one can see, the abnormality was detected before the fault event reported by the operation, which explains the poor FDR and monotonous FAR values obtained by all regressors regardless of the variable selection algorithm.

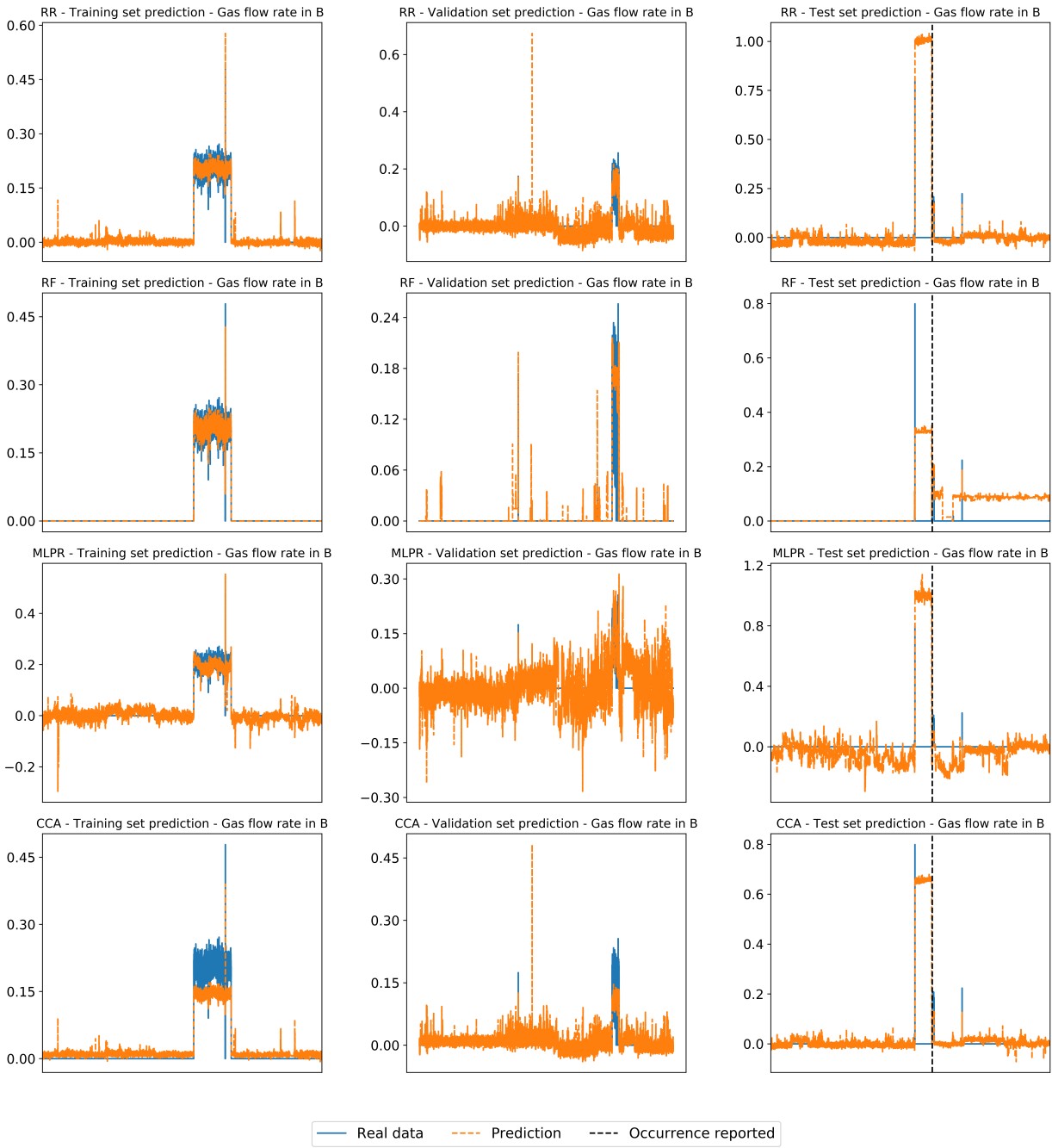

**Figure 2.** Regressors prediction for dimensionless Gas flow rate in fiscal meter 02B (Section A) along Fault-FI when the PCMCI algorithm (partial correlation) was used as the variable selection procedure.

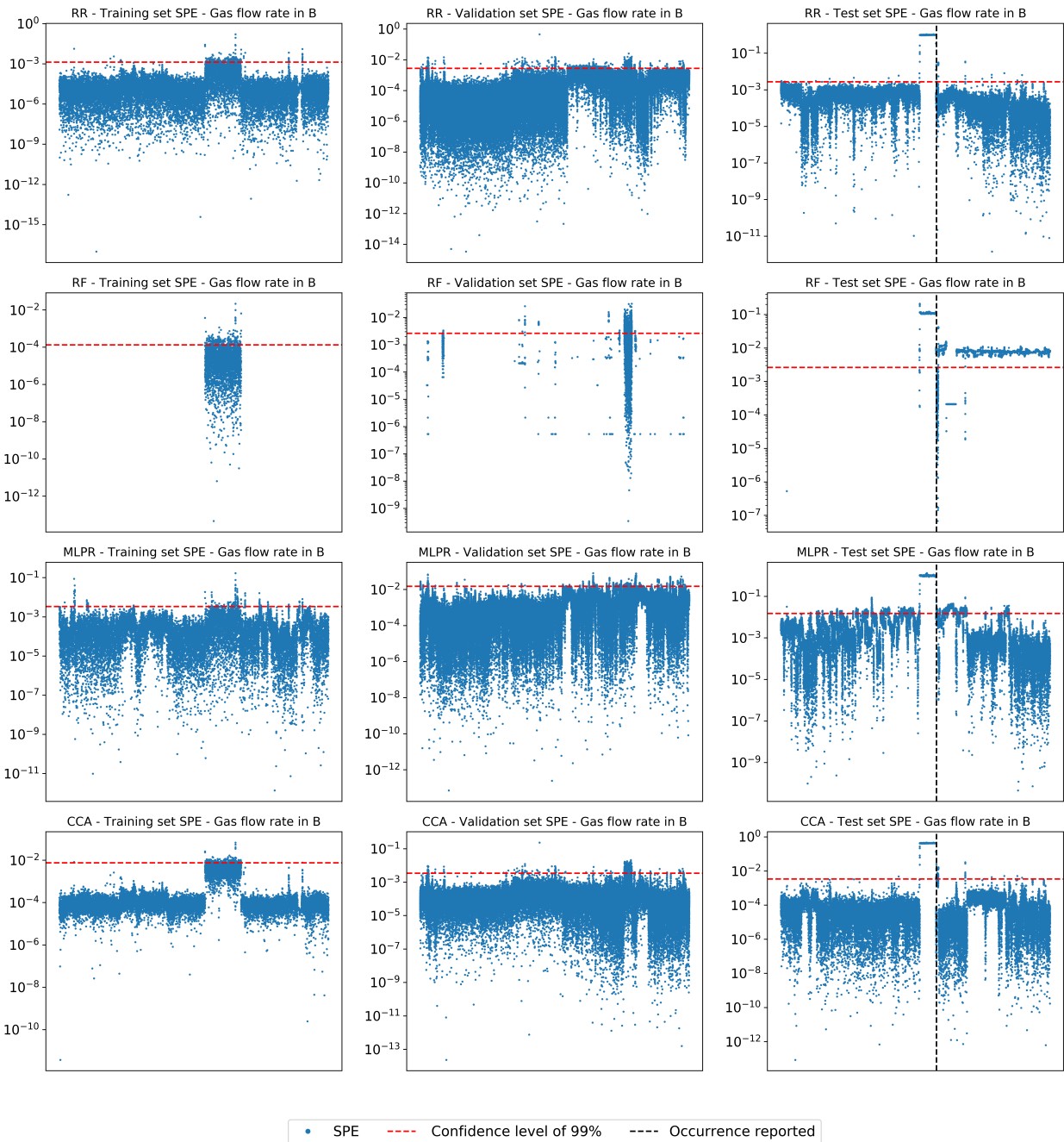

**Figure 3.** Fault detection index SPE along Fault-FI when the PCMCI algorithm (partial correlation) was used as the variable selection procedure.

Figures 4 and 5 show, respectively, the dimensionless temperature predictions and SPE index during Fault II detection. In this case, the PCStable (partial correlation) with MCI (conditional mutual information) algorithm was used as the variable selection procedure. The fault was properly detected according to the reported event and SPE behavior. On the other hand, the intermittent nature of this failure explains the poorer obtained FDR values.

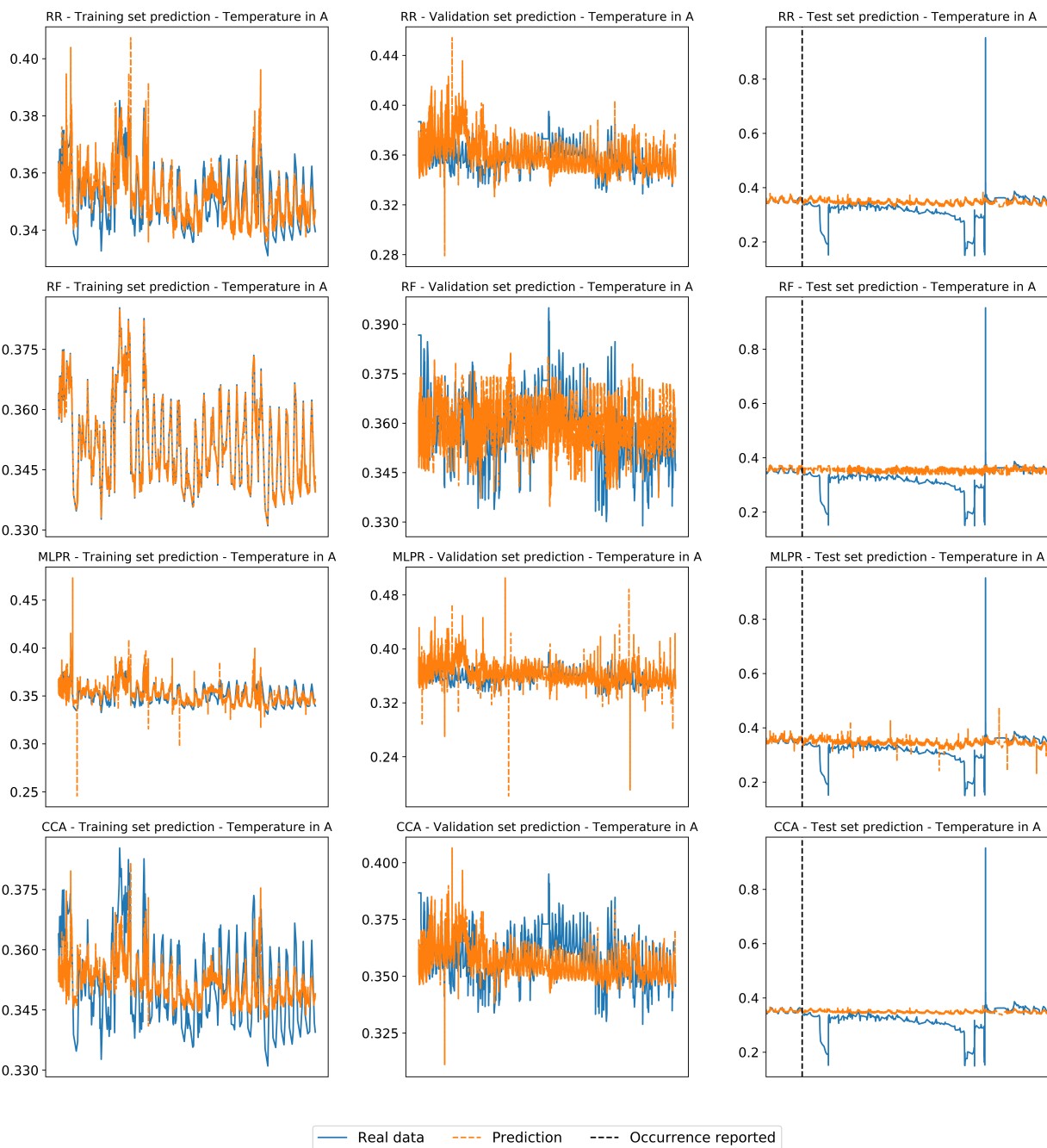

**Figure 4.** Regressors prediction for dimensionless Temperature in fiscal meter 02A (Section A) along Fault-FII when the PCStable algorithm (partial correlation) with MCI (conditional mutual information) was used as the variable selection procedure.

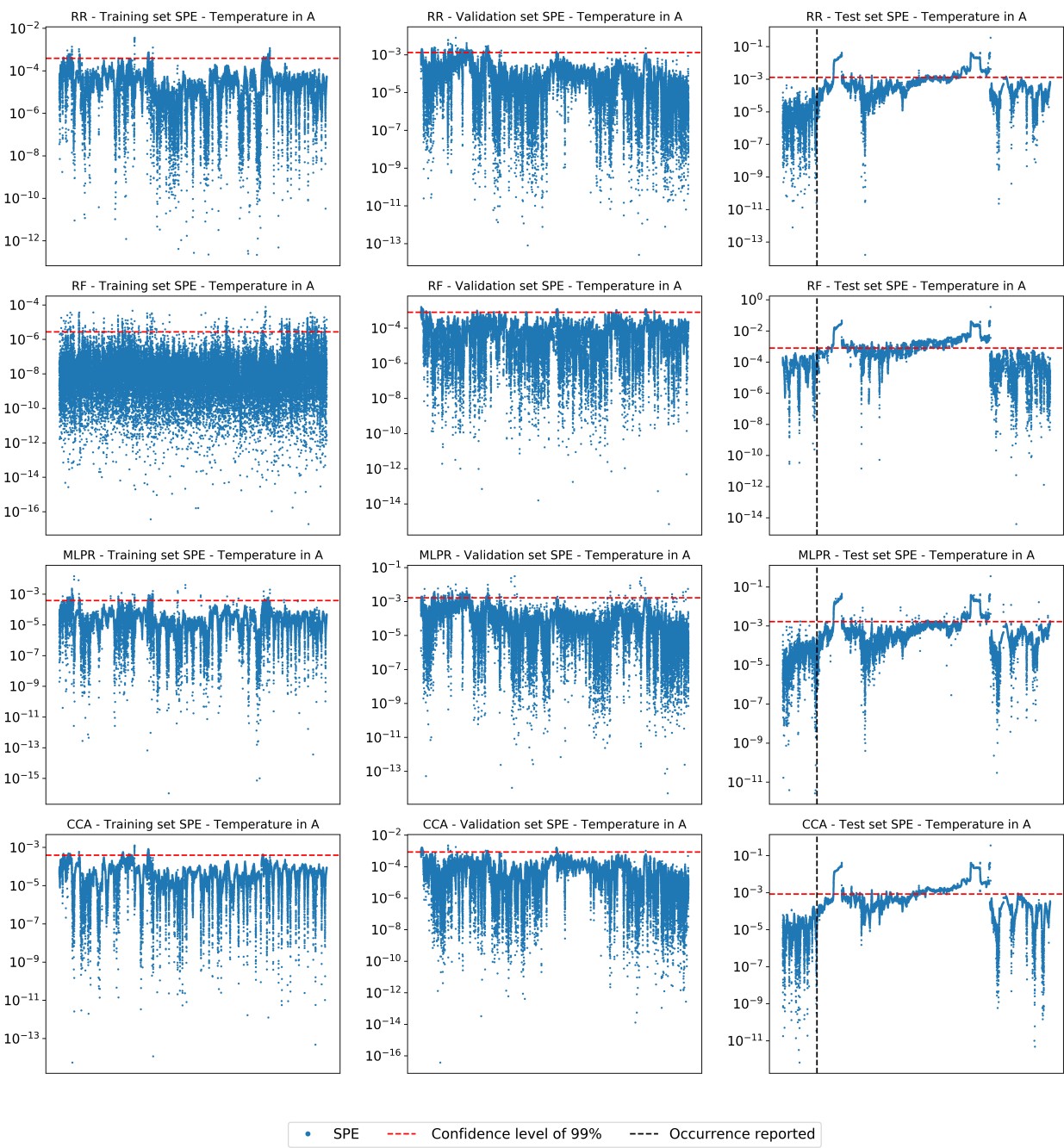

**Figure 5.** Fault detection index SPE along Fault-FII when the PCStable algorithm (partial correlation) with MCI (conditional mutual information) was used as the variable selection procedure.

Finally, the prediction results and SPE index behavior in the Fault-III detection scenario are presented in Figures 6 and 7, respectively. As previously pointed out, this fault was detected appropriately, despite the oscillatory character of the predicted variable. Moreover, the event reported by the operation seemed to have occurred before the actual manifestation of the failure; consequently, the maximum reachable FDR rate corresponds (approximately) to the value of 63% reported in Tables 7–10.

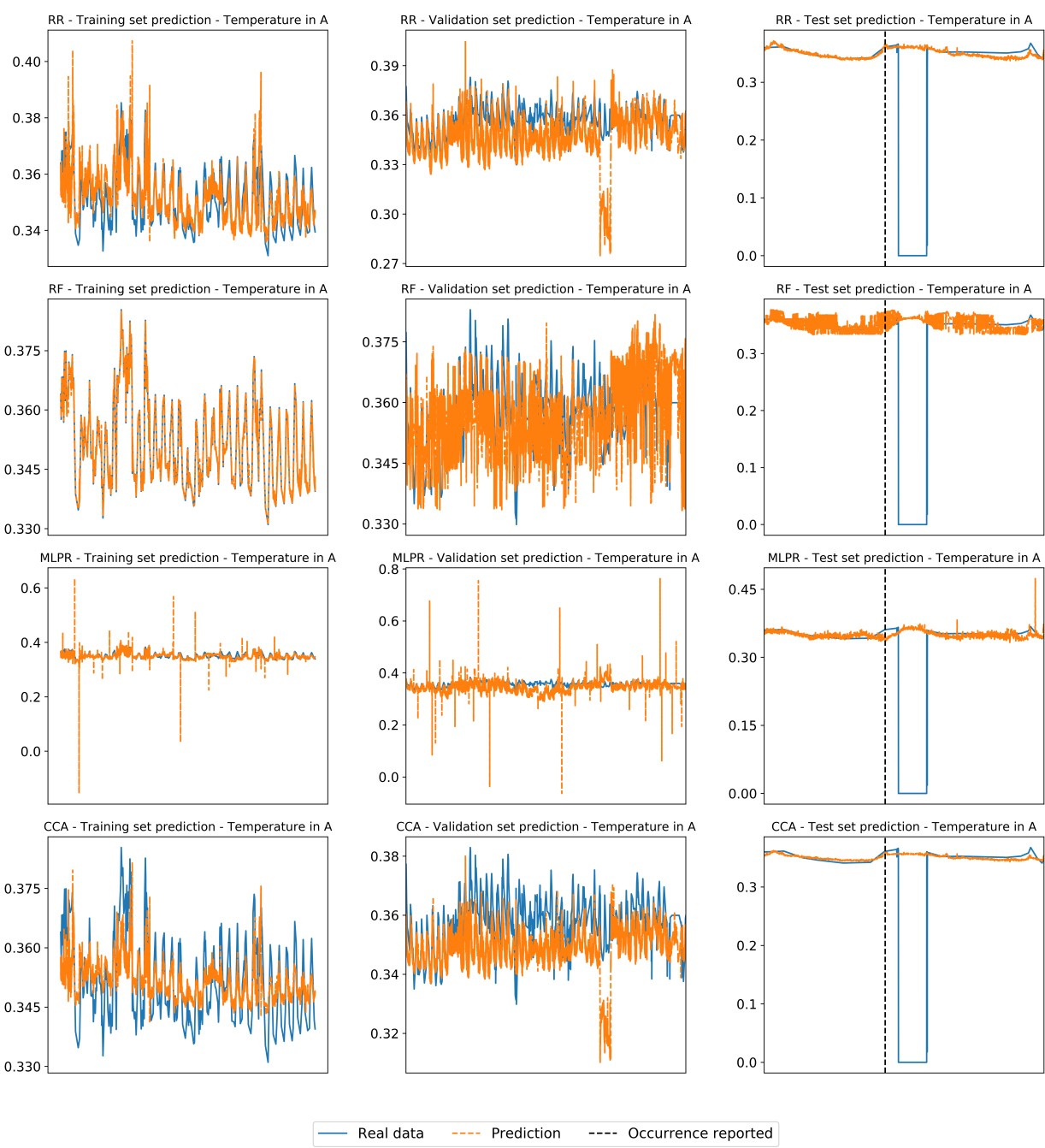

**Figure 6.** Regressors prediction for dimensionless Temperature in fiscal meter 02A (Section A) along Fault-FIII when the PCStable algorithm (partial correlation) with MCI (conditional mutual information) was used as the variable selection procedure.

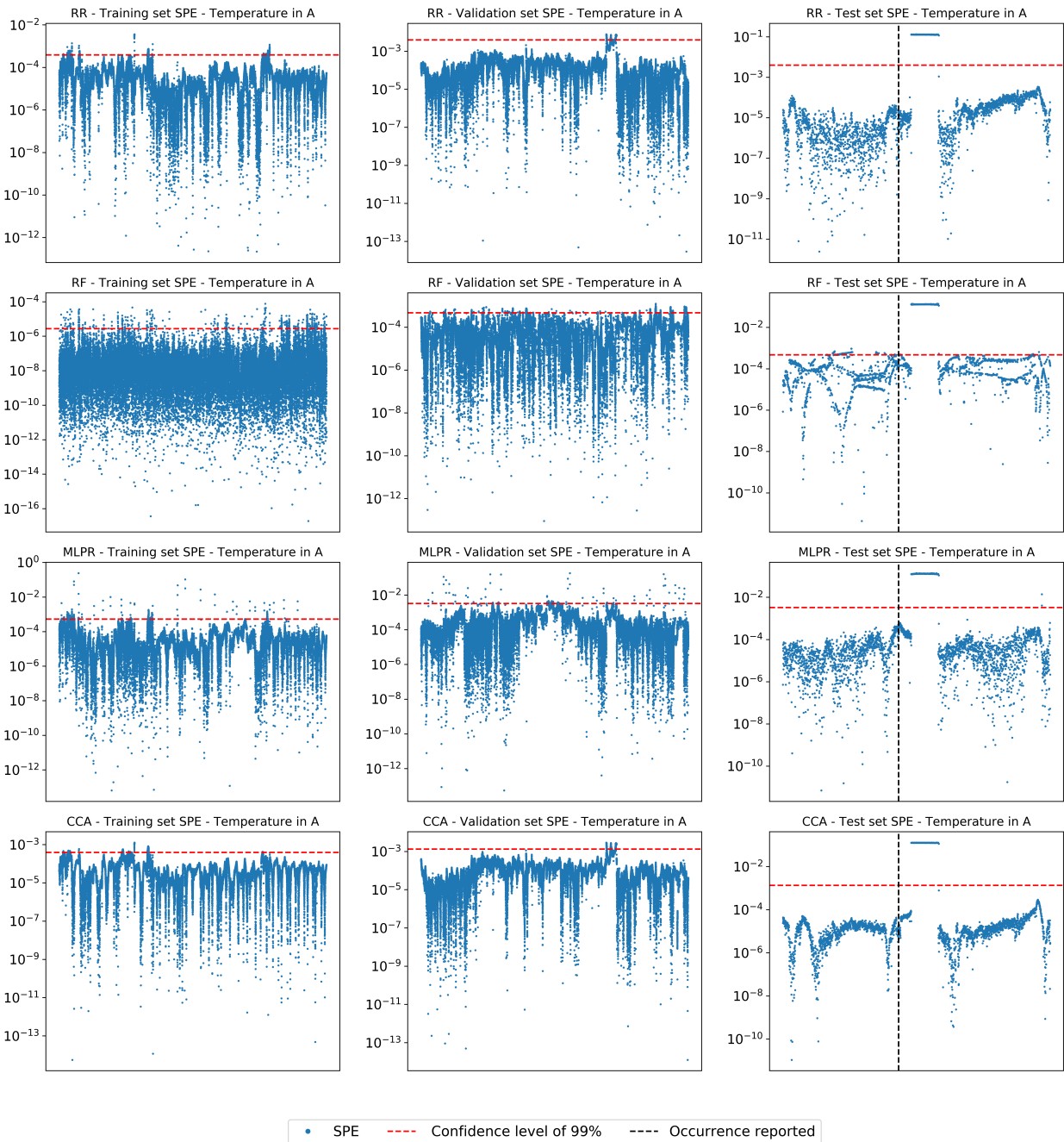

**Figure 7.** Fault detection index SPE along Fault-FIII when the PCStable algorithm (partial correlation) with MCI (conditional mutual information) was used as the variable selection procedure.

An important aspect of the discussion about variable selection methods based on causality is the insertion of lagged variables in the analysis, which derives, naturally, from the discovery and reconstruction of lagged links. The inclusion of these time-shifted variables can allow for improved modelling of the dynamic behaviour of process trajectories, while using the same detection model [56–58].

Mutual information, which was applied in filter methods, is a metric that is similar to those used in causal methods. However, this methodology determines the relationships between pairwise variables, neglecting the effect of the remaining variables on the pair. Therefore, conditional approaches are more appropriate as they attempt to isolate the effects of variables during the discovery of causal connections. Basically, while one approach looks for correlated (nonlinearly) variables, the other approaches look for causal variables.

As previously highlighted, lagged-conditionally independence discovery procedures search for the causal connections of the predicted variable $Y$. Hence, the use of lagged variables seems natural to define the subset of the selected variables.

### 4.2. Performance on Benchmark Case

As described previously, the size of the training subset was kept constant, as determined through PCA analysis, being required 15 components to describe 99.5 % of cumulative variance. The complete PCA analysis is shown in Figure A5 in Appendix A.3.

Table 11 shows the regressors performance when applying the most prominent variable selection procedures by class. According to FDR and FAR metrics, the detection of Fault IDV(1) was better when the PCMCI approach was employed, while Fault IDV(5) was correctly detected with similar performance by the PCMCI and l1-regularization (Lasso) methods. The obtained $R^2$ values in the test sets reflect that they are composed mostly of no-faulty data.

**Table 11.** Performance of the variable selection procedures used to analyze the TEP process.

| Variable Selection Method | Fault | Regressor | FDR (%) | FAR (%) | $R^2$ Training Set | $R^2$ Training Set | $R^2$ Training Set |
|---|---|---|---|---|---|---|---|
| Without variable selection procedure | IDV(1) | RR | 48.63 | 1.25 | 0.35 | 0.28 | 0.77 |
| | | RF | 75.23 | 0.0 | 0.94 | 0.61 | 0.37 |
| | | MLPR | 37.43 | 0.62 | −996.67 | −1162.42 | −332.96 |
| | | CCA | 73.94 | 1.56 | 0.01 | 0.01 | −0.04 |
| | IDV(5) | RR | 99.0 | 0.62 | 0.68 | 0.66 | −152.75 |
| | | RF | 41.99 | 0.0 | 0.91 | 0.65 | 0.55 |
| | | MLPR | 26.93 | 0.62 | −845.53 | −873.52 | −1085.37 |
| | | CCA | 45.31 | 0.62 | 0.01 | 0.01 | −0.09 |
| Mutual information-based | IDV(I) | RR | 44.07 | 0.62 | 0.23 | 0.19 | 0.77 |
| | | RF | 61.49 | 0.0 | 0.98 | 0.58 | 0.31 |
| | | MLPR | 88.46 | 1.25 | −0.78 | −0.82 | −0.54 |
| | | CCA | 61.08 | 1.25 | −0.34 | −0.52 | 0.14 |
| | IDV(5) | RR | 99.0 | 1.25 | 0.60 | 0.6 | −389.82 |
| | | RF | 24.0 | 0.0 | 0.91 | 0.62 | 0.49 |
| | | MLPR | 23.62 | 0.94 | −769.7 | −760.7 | −886.25 |
| | | CCA | 30.75 | 0.94 | 0.06 | 0.06 | −1.06 |
| Forward feature elimination (Lasso) | IDV(I) | RR | 24.52 | 1.56 | 0.32 | 0.28 | 0.84 |
| | | RF | 61.91 | 0.0 | 0.91 | 0.59 | 0.41 |
| | | MLPR | 72.28 | 0.62 | −183.4 | −212.14 | −171.64 |
| | | CCA | 41.99 | 0.94 | 0.24 | 0.18 | 0.72 |
| | IDV(5) | RR | 44.89 | 1.25 | 0.53 | 0.52 | 0.71 |
| | | RF | 44.75 | 0.0 | 0.91 | 0.62 | 0.61 |
| | | MLPR | 29.22 | 0.62 | −201.32 | −205.56 | −110.87 |
| | | CCA | 56.17 | 0.94 | 0.37 | 0.37 | 0.44 |

**Table 11.** *Cont.*

| Variable Selection Method | Fault | Regressor | FDR (%) | FAR (%) | $R^2$ Training Set | $R^2$ Training Set | $R^2$ Training Set |
|---|---|---|---|---|---|---|---|
| L1-regularization (Lasso) | IDV(I) | RR | 32.06 | 1.25 | 0.33 | 0.28 | 0.87 |
| | | RF | 84.54 | 0.0 | 0.93 | 0.64 | 0.32 |
| | | MLPR | 97.30 | 1.56 | −27.59 | −30.59 | −103.89 |
| | | CCA | 83.12 | 0.94 | 0.06 | -0.04 | 0.44 |
| | IDV(5) | RR | 46.95 | 0.94 | 0.54 | 0.54 | 0.64 |
| | | RF | 61.84 | 0.0 | 0.92 | 0.65 | 0.53 |
| | | MLPR | 80.99 | 2.19 | −99.74 | −100.12 | −360.55 |
| | | CCA | 61.13 | 0.31 | 0.39 | 0.38 | 0.39 |
| PCStable (Partial correlation) + MCI (Conditional mutual information) | IDV(I) | RR | 32.06 | 1.25 | 0.33 | 0.28 | 0.87 |
| | | RF | 84.54 | 0.0 | 0.93 | 0.60 | 0.32 |
| | | MLPR | 97.30 | 1.56 | −27.59 | −30.59 | −103.89 |
| | | CCA | 83.12 | 0.94 | 0.06 | −0.04 | 0.44 |
| | IDV(5) | RR | 46.95 | 0.94 | 0.54 | 0.54 | 0.64 |
| | | RF | 61.84 | 0.0 | 0.92 | 0.65 | 0.53 |
| | | MLPR | 80.99 | 2.19 | −99.74 | −100.12 | −360.55 |
| | | CCA | 61.13 | 0.31 | 0.39 | 0.38 | 0.39 |

The performance metrics corresponds to fault detection rate (FDR%), false alarm rate (FAR%) and regression score $R^2$.

The better performance of the causal methods for variable selection in this case study can be explained by the inclusion of lagged variables for model training, which according to the literature [59,60], can exert a determining role in the detection of failures in the TEP process.

It is worth mentioning that the use of variable selection methods (except the causal methods) did not lead to notable improvements in relation to the reference performance. Hence, the use of variable selection schemes in TEP case study does not constitute a limiting step for detection of the analyzed faults, as the process variables are more causally interconnected and the redundant variables do not interfere drastically in the performance of the models. However, the selection of variables allows working with less complex and computationally faster models. Moreover, it must be clear that the use of causal methods for selection of relevant variables did allow the improvement of the analyzed performance, being recommended for more involving implementations.

### 4.3. Analysis of Selected Variables

The oil and gas fiscal metering process constitutes an interesting case study because it involves a large number of variables measured along the different sections of the process, making it difficult to define a priori the most relevant variables for the prediction of a particular variable of interest. Intuitively, it is expected that this subset will contain variables from the same plant section to which the prediction target variable belongs and reflects phenomenological characteristics of the process. In this context, Figures A9–A12 in Appendix A.5 show the subsets selected by the most outstanding selection methods (by class) according to the previously reported results. These selections correspond to the training set used to detect Fault F-I, where the predicted variable corresponded to FIT-02B-A (gas flow rate in fiscal meter 2B in Section A of Figure 1). The process variables and respective tags are listed in Table A3 in Appendix A.6.

The ranking of relevant variables determined by the distinct variable selection methods show PDIT02B-A (differential pressure in fiscal meter 2B in Section A in Figure 1) as most

important measurement, which is consistent with the inherent physical principle of the fiscal meter measurement. However, it was the causal methods that considered in their respective selected subsets the largest amount of variables geographically adjacent to the monitored fiscal meter, representing the phenomenological nature of the process.

On the other hand, in systems of high dimensionality, the causal characterization methods are useful not only for fault diagnosis ([61–63]), but also for generating better models for fault detection as already shown in this work. In addition, the causal networks reconstructed from time series [36] keep some causal properties that can be intuitively extracted from the respective process flow diagram (PFD).

Another representative performance metrics is the mean absolute error (MAE). Figure 8 shows the MAE obtained by the different regressors for Fault F-I in the validation set considering all the variable selection methods studied here. As one can see, the MAE values were lower when the methods for selecting variables based on causality were used. It is important to note that better adjustments and performance can be possibly achieved if hyperparameters optimization stages are carried out during the training procedures. However, as the present work emphasized the study of the effect of the variable selection procedures and not of the effect of hyperparameters on the regression model performances during fault detection, optimization of hyperparameters was not sought.

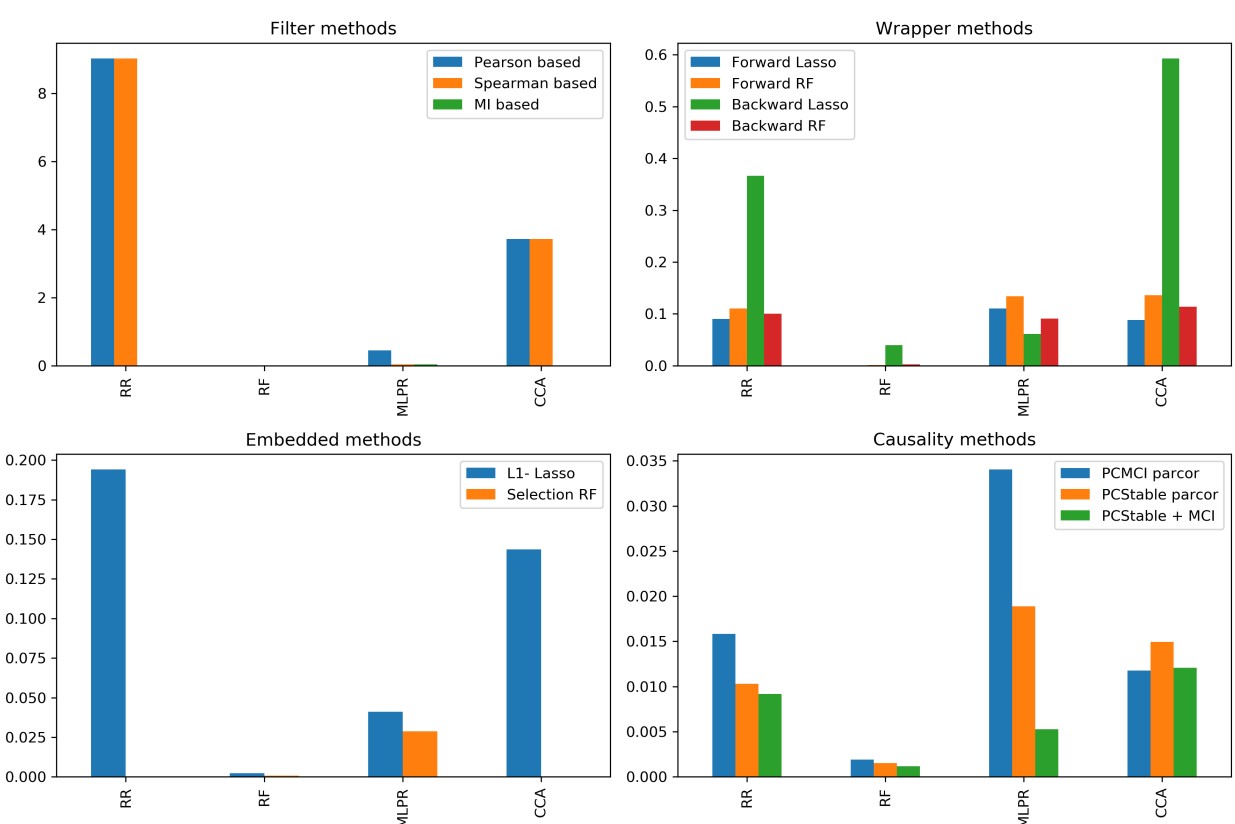

**Figure 8.** MAE in validation set considering all variable selection methods applied in Fault FI.

Finally, Table 12 shows the CPU times demanded by each method during the selection of variables for the detection of Fault F-I. It can be observed that the causal method were the slowest ones, given the more involving computation of causal links. However, considering that the variable selection stage must be performed before the training stage, this computational demand would not constitute a limiting factor for eventual online applications.

**Table 12.** CPU times demanded for the variable selection methods that provided the best performance in each class for detection of Faul F-I in the industrial case.

| Variable Selection Method | Class | CPU Time (s) |
|---|---|---|
| Mutual information-based | Filter | 138 |
| Forward feature elimination (Lasso) | Wrapper | 720 |
| L1-regularization (Lasso) | Embedded | 43 |
| PCMCI (Partial correlation) | Filter (causal) | 1403 |

*4.4. Final Considerations*

In general, all fault detection metrics showed improvements when applied any variable selection approach studied in this work. Moreover, these approaches reduced the fault detection problem dimensionality, allowing building simple learning models in which is a desired attribute in online monitoring.

Variable selection methods based on causality led to better performance in fault detection since included lagged-time variables addressed to model the dynamic behavior of the process trajectories. Furthermore, as was discussed in Section 4.3, the selected variables subset kept causal associations in respect to the predicted variable reflecting phenomenological characteristics of the process.

The results obtained showed that the wrapper-based methods prevail over filter-based methods in terms of prediction accuracy, as similarly observed in the literature [3,6]. However, causality methods can be classified as filter-based methods because the variable selection engine is independent of the regressor model. This independence explains the homogeneity in terms of fault detection metrics observed in the four learning models along the faults scenarios studied.

The fault detection scenarios corresponding to the real industrial case provided the opportunity to work with issues rarely found in simulated or benchmark cases such as high dimensionality, real noised measures, and divergences between the fault events reported and the actual manifestation of the failure.

**5. Conclusions**

In the present paper, variable selection methods based on causality are implemented, analyzed and then obtained performances were compared with the performances obtained with several other filter-based, wrapper-based, and embedded-based variable selection methods. Two case studies were presented, corresponding to a simulated benchmark (the Tennessee-Eastman process) and an actual industrial case (a fiscal gas metering station). As shown through many examples, all learning algorithms considered in the present work provided better regression and fault detection performances when using variable selection procedures based on causality. In particular, the variable selection approaches based on causality establish the causal connections of the predicted variable, also allowing the determination of the respective lagged-conditionally dependence. Hence, the subset of the selected variables reflects phenomenological characteristics of the process, as it became evident in the industrial case study. Although the variable selection methods based on causality were more computationally demanding, the use of these methods in monitoring scenarios that involve a large number of variables is highly recommended, especially because it can be performed as pre-processing data analysis stage and thus does not compromise the characteristic computation time of the final application.

Let us also discuss some directions for future research. In the present work, we proposed the use of the causal discovery approach in variable selection methods addressed to assists the process fault detection. As discussed above, these causal methods are based on the estimation of lagged-conditional dependence of the measured variables. In high-dimensional systems, estimating these dependencies involve the computation of complicated density probability function, which can lead to inaccurate or spurious estima-

tions of the causal relationships [37]. Future work should include the use of ruled-based frameworks belonging to expert knowledge [64] as equipment adjacency in the process plant, aiming at the incorporation of physical and operational restrictions in the estimation of the causal relationships. Finally, it would be useful to propose other benchmark cases in order to test these causal methods in feature/variable selection problems with high-redundancy datasets, aiming to evaluate the approach robustness in the presence of correlated measures.

**Author Contributions:** Conceptualization, M.M.C., T.K.A., and J.C.P.; data curation, A.M., M.M.C., L.F.d.O.C., F.C.D., T.K.A., and P.H.T.; funding acquisition, T.K.A., F.C.D. and J.C.P.; methodology, T.L.; project administration, T.L., L.F.d.O.C., T.K.A., F.C.D., and P.H.T.; resources, T.K.A., F.C.D., P.H.T., and F.C.D.; software, A.M., M.M.C., R.M.S., and P.H.T.; supervision, J.C.P.; writing, original draft, N.C., M.M.C., and T.K.A.; writing, review and editing, N.C., M.M.C., R.M.S., J.C.P. and T.K.A. All authors have read and agreed to the published version of the manuscript.

**Funding:** This study was financed in part by the Coordenação de Aperfeiçoamento de Pessoal de Nível Superior-Brasil (CAPES)-Finance Code 001. The authors also thank CNPq-Conselho Nacional de Desenvolvimento Científico e Tecnológico, and Petrobras (Petróleo Brasileiro SA), for the financial support of this work, as well as for covering the costs to publish in open access.

**Institutional Review Board Statement:** Not applicable.

**Informed Consent Statement:** Not applicable.

**Data Availability Statement:** Data sharing is not applicable.

**Conflicts of Interest:** The authors declare no conflict of interest. The founders had no role in the design of the study; in the collection, analyses, or interpretation of data; in the writing of the manuscript; nor in the decision to publish the results.

## Abbreviations

The following abbreviations are used in this manuscript:

| | |
|---|---|
| MI | Mutual information |
| JMI | Joint mutual information |
| CMI | Conditional mutual information |
| DMI | Dynamic mutual information |
| TE | Transfer entropy |
| PDF | Probability density function |
| DAG | Directed acyclic graph |
| TEP | Tennessee Eastman process |
| SPE | Square prediction error |
| PCA | Principal components analysis |
| FDR | Fault detection rate |
| FAR | False alarm rate |
| RF | Random Forest |
| RR | Ridge regression |
| MLPR | Multi-layer perceptron regressor |
| CCA | Canonical correlation analysis |
| MCI | Mutual conditional independence |
| PFD | Process flow diagram |
| MAE | Mean absolute error |

## Appendix A

*Appendix A.1. PC Algorithm and PC-Stable Algorithm*

Given a Directed Acyclic Graph (DAG) $G$ consisting of a set of nodes or vertices $V$ (circles) and a set of edges (lines) $E \subseteq V \times V$. A set of variables and its causal interactions can be represented, respectively, by the nodes and edges of a DAG, where the direction and measure of these interactions are obtained according to described in Section 2.3.

The PC algorithms are based on the conditional independence evaluation that establishes the existence of a link (edge) between the variables (nodes) $X$ and $Y$, if given a set of conditions $Z$, $X$ and $Y$ are not independent conditioning on $Z$.

An intuitive approach to construct the complete network map (DAG) consists of the exhaustive search of all possible conditions $Z$ to determine if $X$ and $Y$ are conditionally connected. However, this is a computationally inefficient method, turning the PC algorithm or PC-Stable algorithm as interesting approaches to address causal link characterization problems in high dimensional systems.

The PC Algorithm considers, at the beginning, a network fully connected. For each edge, tests if the pair of variables connected, $X$ and $Y$, are independent conditioning on a subset $Z$ of all neighbours/parents of $X$ and $Y$ and remove or retain the respective edge based on the result. The mutual conditional independence tests (MCI) are applied by levels, according to the size $d$ of the conditions set $Z$. At the first level ($d = 0$), all pairs of nodes (variables) are tested, conditioning on the empty set $Z$. The algorithm can remove some of the edges (links) and only tests the remaining edges in the next level ($d = 1$). The size of the conditioning set, $d$, is progressively increased until $d$ is greater than the size of the adjacent sets of the testing nodes.

Figure A1 shows an example of PC algorithm being applied to a hypothetical dataset with four nodes, $A$; $B$; $C$; and $D$ [40]. As one can see, three edges remains after level 1 tests (i.e., $Z = []$). At the next level, each remaining edge will be tested conditioned on each neighbour/parent of the testing variables (i.e., $d = 1$). For example, given the edge $A - B$, there are, at most, two tests which are conditioning on $C$ and conditioning on $D$. In particular, the test $MCI(A, B|C)$ returns conditional independence, then the edge is removed from the graph and the algorithm continues testing on the other edge.

Conditional independence test is likely to present inaccurate values in high dimensional systems. Furthermore, for the PC Algorithm, removing or retaining an edge would result in changes in the condition set $Z$ of other nodes since the network graph is updated dynamically. Therefore, the resulting network graph is dependent on the order in which the conditional independence tests are performed [40]. For example, in Figure A1, when the test $MCI(A, B|C)$ returns conditional independence and, consequently, the edge is removed, the set of neighbors/parents of $A$ is also updated $adj(A) = C, D$. Therefore, when testing the edge $A - C$, the conditions set not contains $B$. Considering the case where the conditional test $MCI(A, B|C)$ wrongly removes the edge $A - C$, it misses the test $MCI(A, C|B)$ which may remove the edge $A - C$. On the other hand, if the procedure tests $MCI(A, C|B)$ first and removes the edge $A - C$, it would end up with a different network.

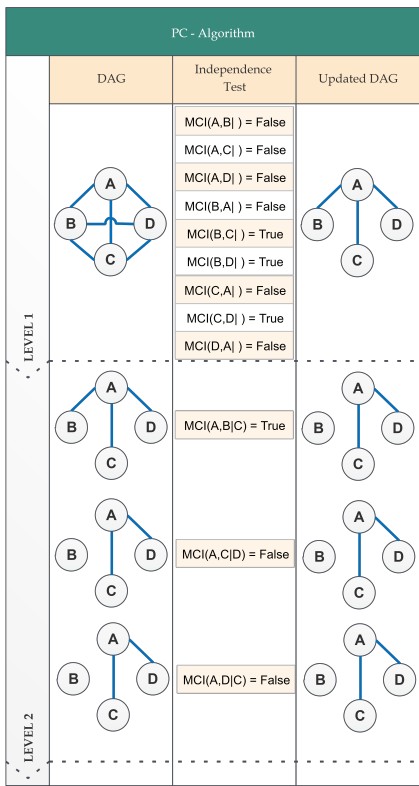

**Figure A1.** Example of applying PC algorithm in a hypothetical dataset [40].

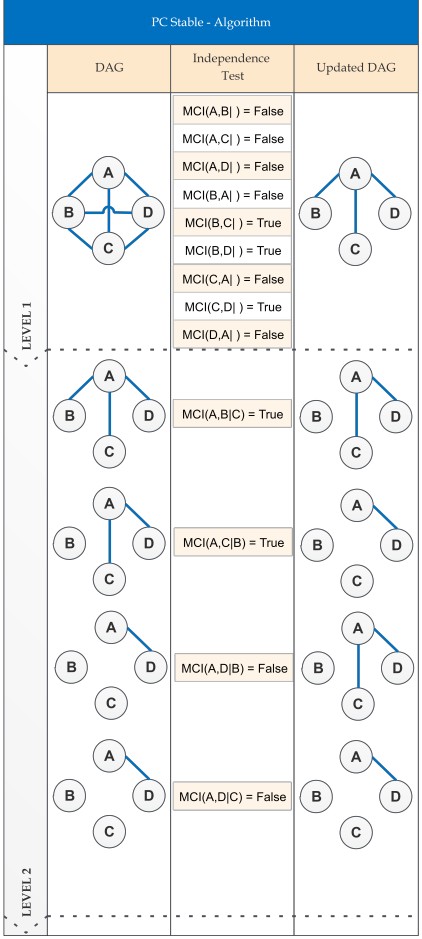

**Figure A2.** Example of applying PC-Stable algorithm in a hypothetical dataset [40].

*Appendix A.2. Tennessee Eastman Process*

Colombo andMaathuis [39] proposed the PC-Stable algorithm addressed to obtain a stable output skeleton that does not depend on how variables are ordered in the input dataset. In this method, the neighbour/parents (adjacent) sets of all nodes are evaluated and kept unchanged at each particular level, preventing that the edge deletion affects the conditioning set of the other nodes. Figure A2 shows the respective network update in each level when the PC-Stable algorithm is applied.

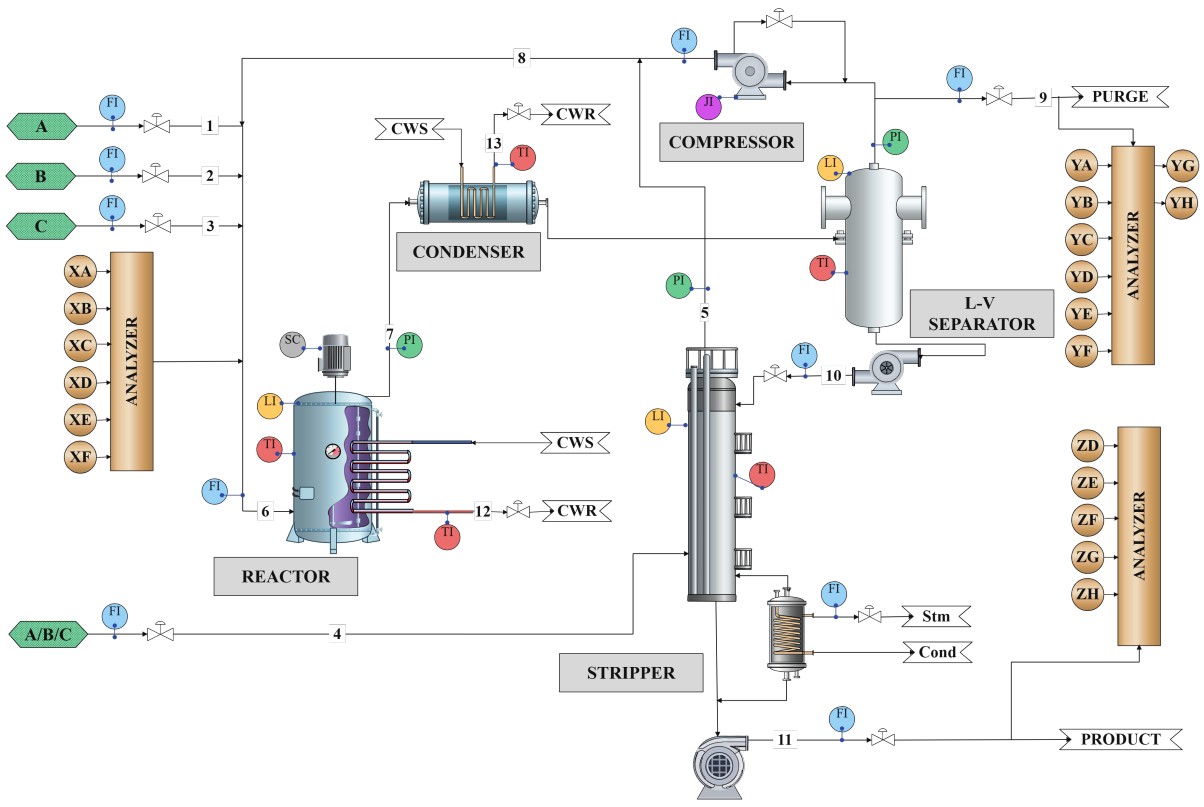

**Figure A3.** A schematic diagram of TEP.

**Table A1.** Measured variables of TEP.

| Measured Variable ID | Description |
| --- | --- |
| F1 | Feed flow component A (stream 1) in kscmh |
| F2 | Feed flow component D (stream 2) in kg/h |
| F3 | Feed flow component E (stream 3) in kg/h |
| F4 | Feed flow components A/B/C (stream 4) in kscmh |
| F5 | Recycle flow to reactor from separator (stream 8) in kscmh |
| F6 | Reactor feed rate (stream 6) in kscmh |
| P7 | Reactor pressure in kPa gauge |
| L8 | Reactor level |
| T9 | Reactor temperature in °C |
| F10 | Purge flow rate (stream 9) in kscmh |
| T11 | Separator temperature in °C |

**Table A1.** *Cont.*

| Measured Variable ID | Description |
|:---:|:---:|
| L12 | Separator level |
| P13 | Separator pressure in kPa gauge |
| F14 | Separator underflow in liquid phase (stream 10) in m³/h |
| L15 | Stripper level |
| P16 | Stripper pressure in kPa gauge |
| F17 | Stripper underflow (stream 11) in m³/h |
| T18 | Stripper temperature in °C |
| F19 | Stripper steam flow in kg/h |
| J20 | Compressor work in kW |
| T21 | Reactor cooling water outlet temperature in °C |
| T22 | Condenser cooling water outlet temperature in °C |
| XA | Concentration of A in reactor feed (stream 6) in mol% |
| XB | Concentration of B in reactor feed (stream 6) in mol% |
| XC | Concentration of C in reactor feed (stream 6) in mol% |
| XD | Concentration of D in reactor feed (stream 6) in mol% |
| XE | Concentration of E in reactor feed (stream 6) in mol% |
| XF | Concentration of F in reactor feed (stream 6) in mol% |
| YA | Concentration of A in purge (stream 9) in mol% |
| YB | Concentration of B in purge (stream 9) in mol% |
| YC | Concentration of C in purge (stream 9) in mol% |
| YD | Concentration of D in purge (stream 9) in mol% |
| YE | Concentration of E in purge (stream 9) in mol% |
| YF | Concentration of F in purge (stream 9) in mol% |
| YG | Concentration of G in purge (stream 9) in mol% |
| YH | Concentration of H in purge (stream 9) in mol% |
| ZD | Concentration of D in stripper underflow (stream 11) in mol% |
| ZE | Concentration of E in stripper underflow (stream 11) in mol% |
| ZF | Concentration of F in stripper underflow (stream 11) in mol% |
| ZG | Concentration of G in stripper underflow (stream 11) in mol% |
| ZH | Concentration of H in stripper underflow (stream 11) in mol% |

**Table A2.** Manipulated variables of TEP.

| Manipulated Variable ID | Description |
|:---:|:---:|
| MV1 | Valve position feed component D (stream 2) |
| MV2 | Valve position feed component E (stream 3) |
| MV3 | Valve position feed component A (stream 1) |
| MV4 | Valve position feed components A/B/C (stream 4) |
| MV5 | Valve position compressor recycle |
| MV6 | Purge valve position (stream 9) |

**Table A2.** *Cont.*

| Manipulated Variable ID | Description |
|---|---|
| MV7 | Valve position underflow separator (stream 10) |
| MV8 | Valve position underflow stripper (stream 11) |
| MV9 | Valve position stripper steam |
| MV10 | Valve position cooling water outlet of reactor |
| MV11 | Valve position cooling water outlet of separator |
| MV12 | Rotation speed of reactor agitator |

*Appendix A.3. Principal Component Analysis in Case Studies*

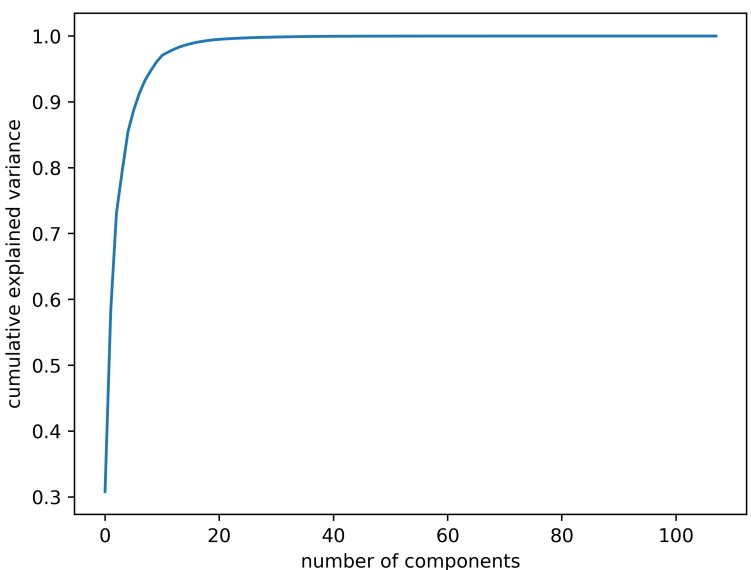

**Figure A4.** Principal component analysis in real industrial dataset.

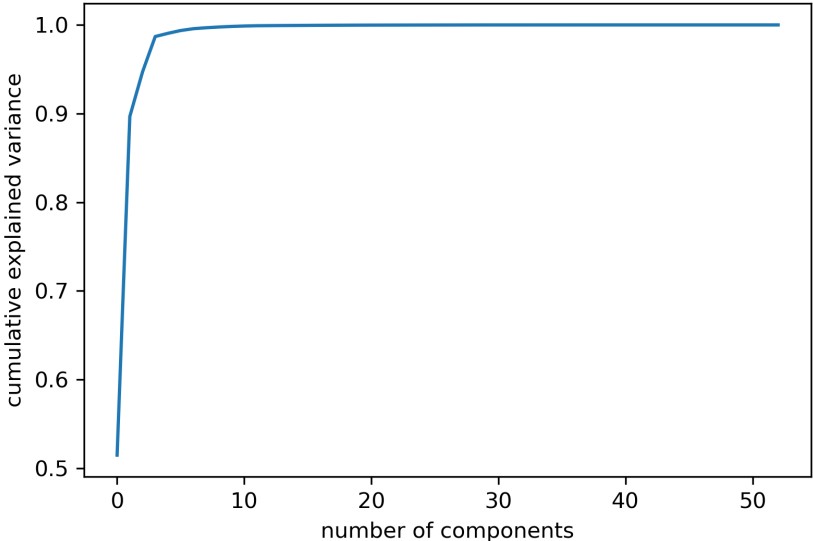

**Figure A5.** Principal component analysis in TEP.

*Appendix A.4. Regressors Prediction of Reference Scenarios in Real Industrial Case*

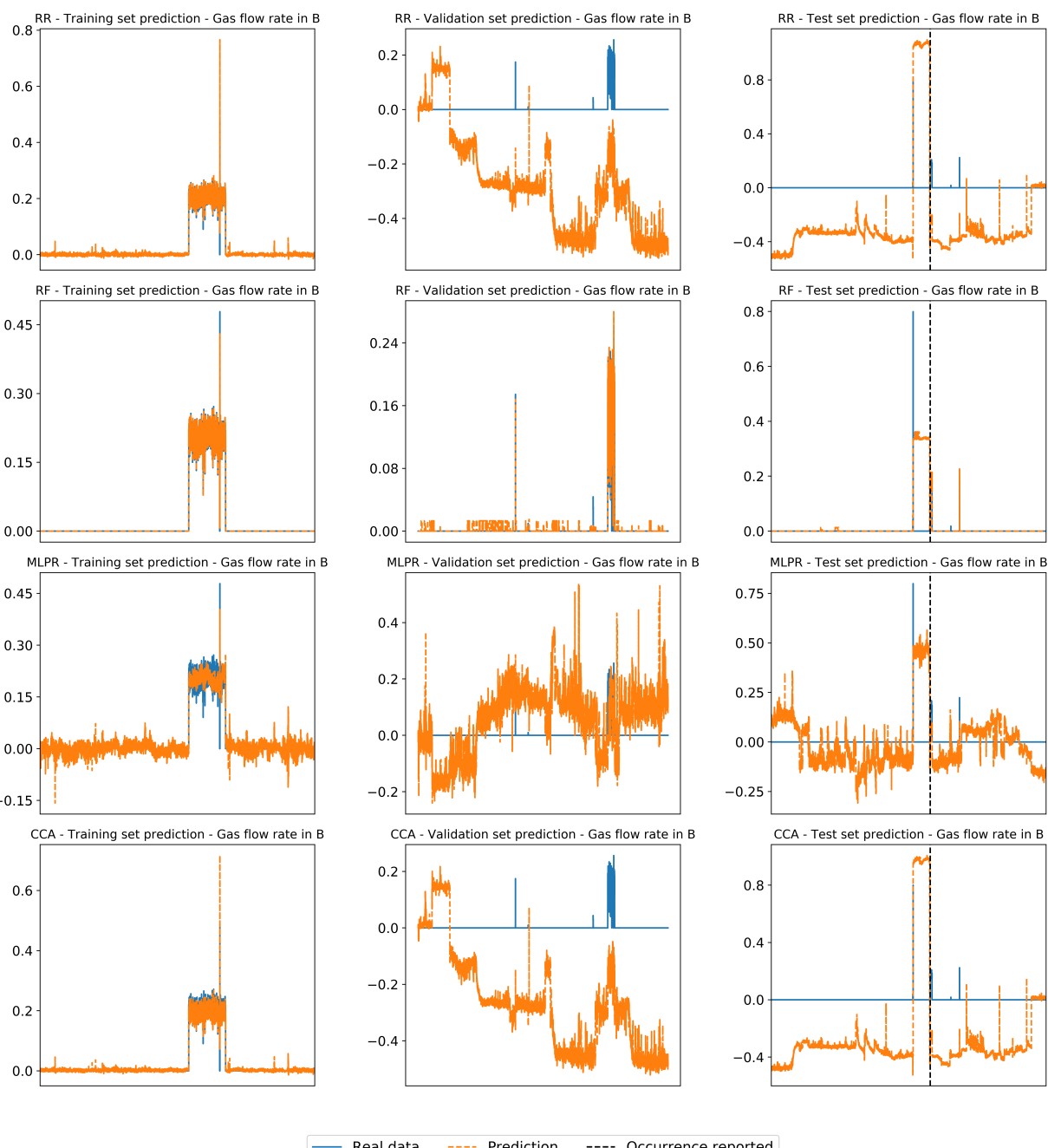

**Figure A6.** Regressors prediction for dimensionless Gas flow rate in fiscal meter 02B (Section A) along Fault-FI without considering variable selection stage.

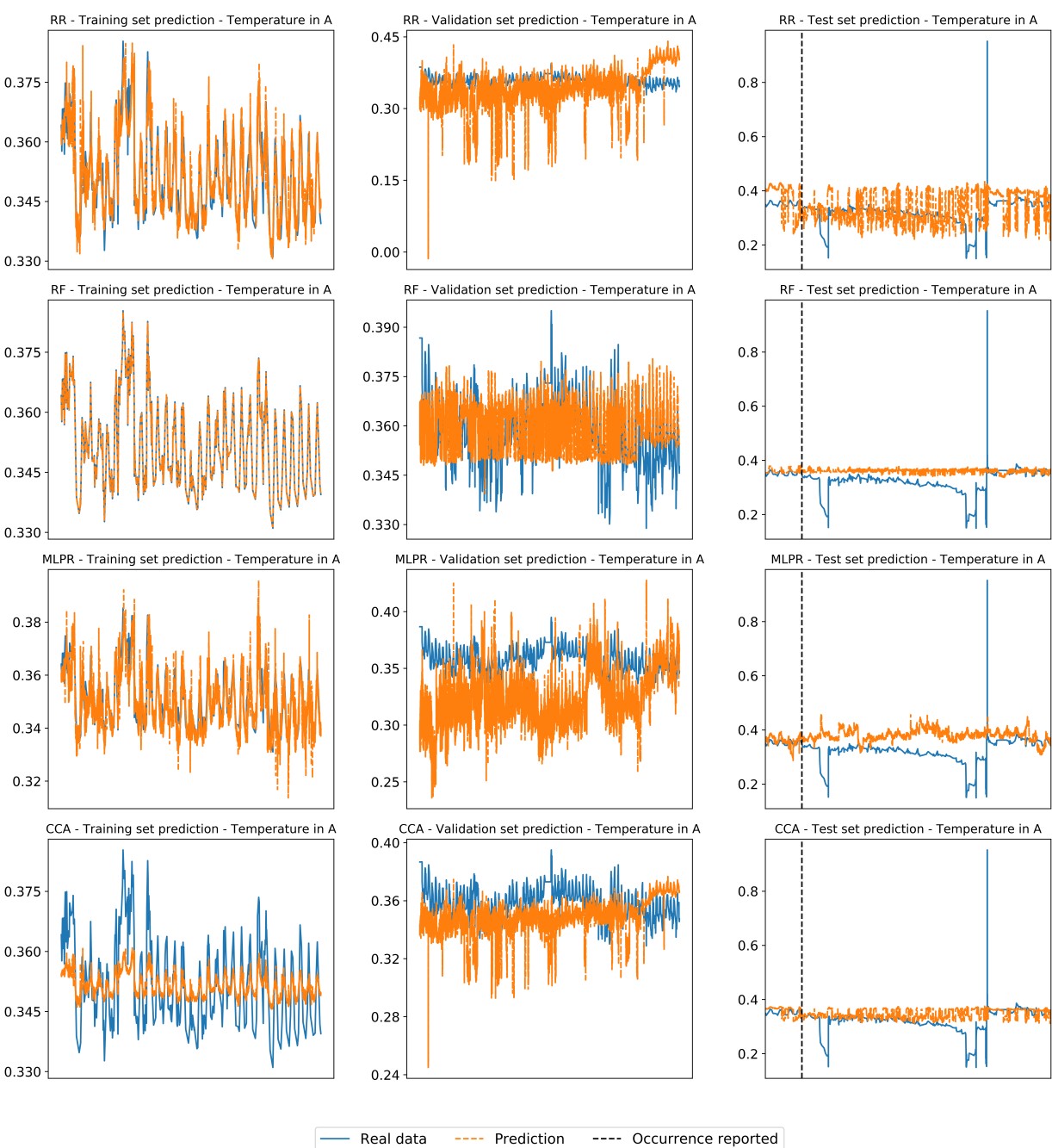

**Figure A7.** Regressors prediction for dimensionless temperature in fiscal meter 02A (Section A) along Fault-FII without considering variable selection stage.

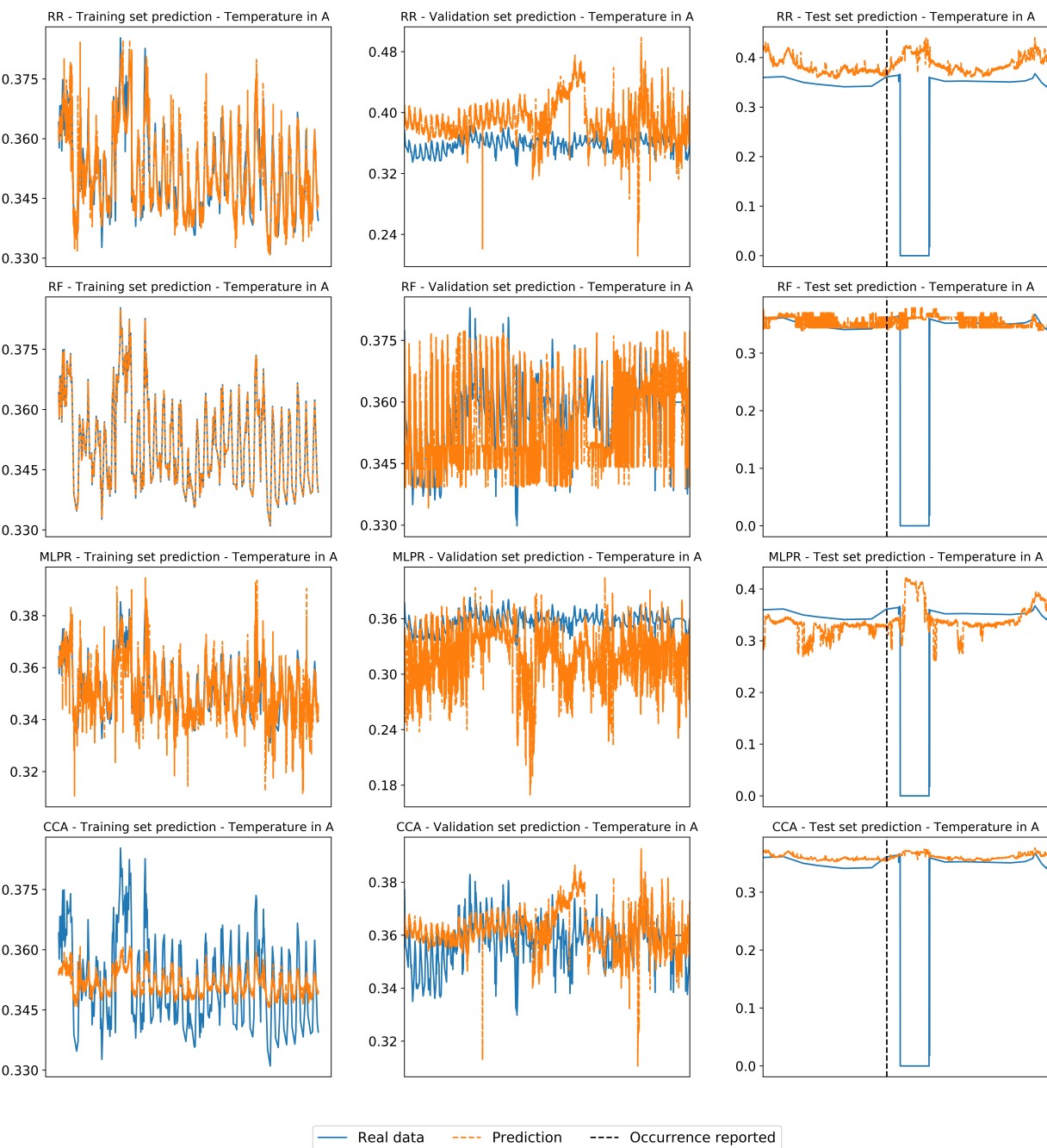

**Figure A8.** Regressors prediction for dimensionless temperature in fiscal meter 02A (Section A) along Fault-FIII without considering variable selection stage.

*Appendix A.5. Selected Subsets in Fault Detection F-I Scenario*

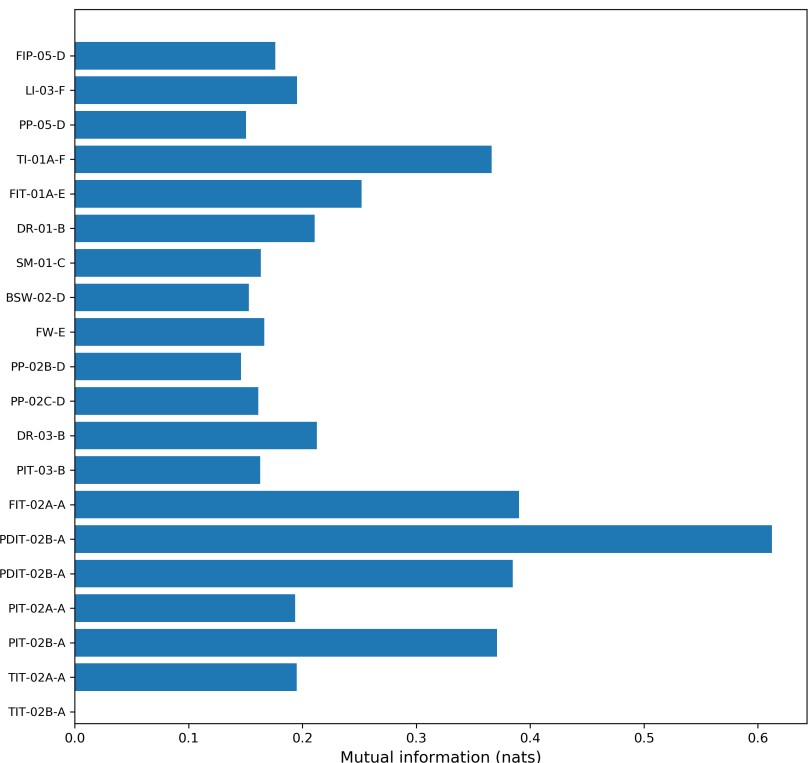

**Figure A9.** Variables subset for Fault F-I detection selected by the filter method based on mutual information.

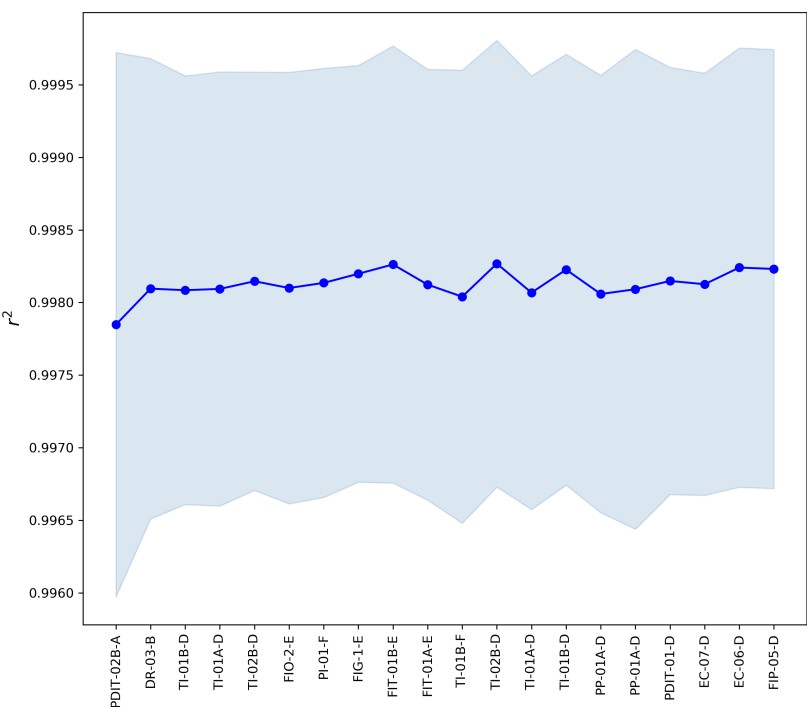

**Figure A10.** Variables subset for Fault F-I detection selected by the forward feature elimination (Lasso) method.

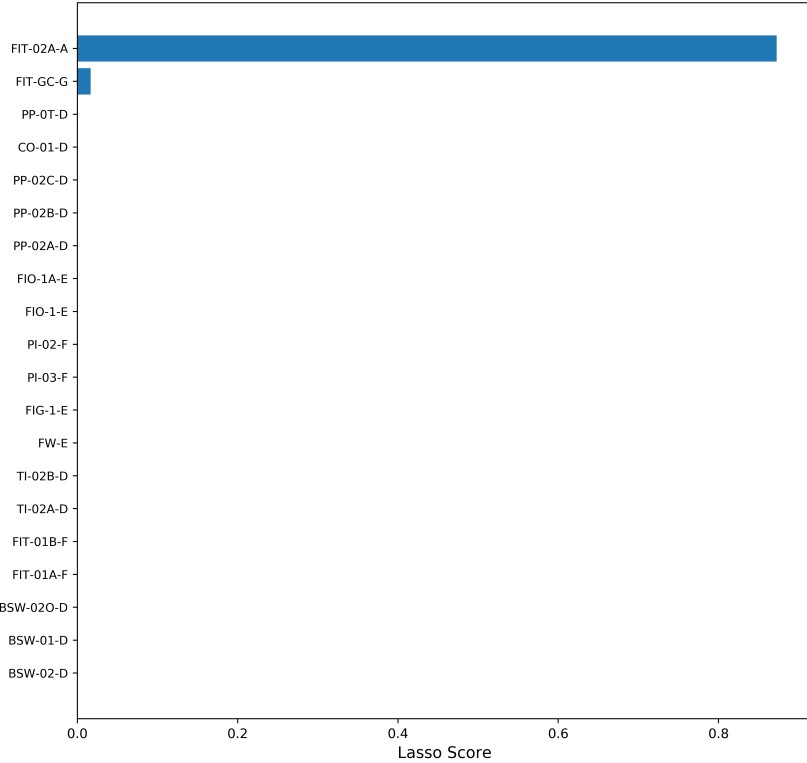

**Figure A11.** Variables subset for Fault F-I detection selected by the L1-regularization (Lasso) method.

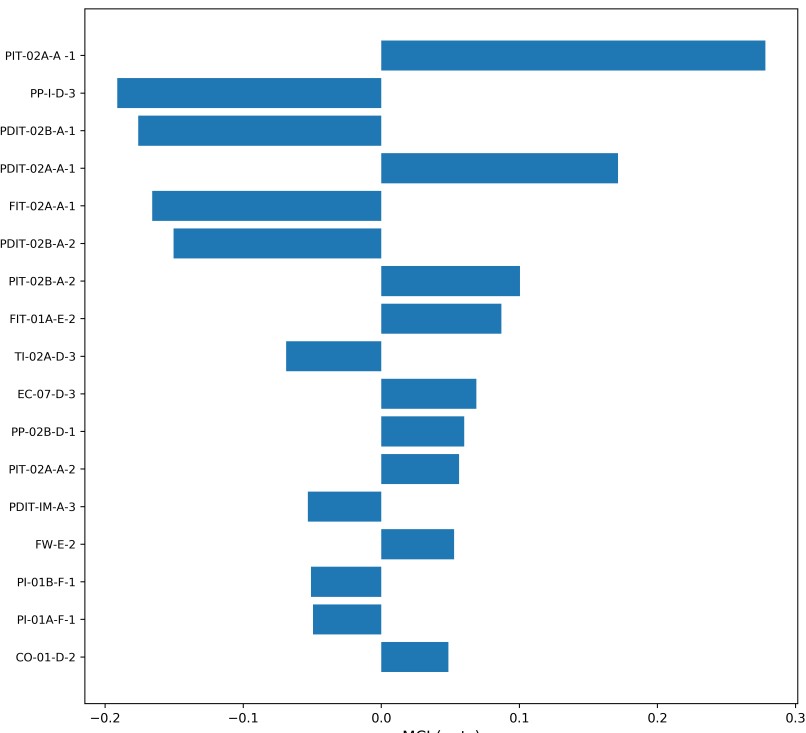

**Figure A12.** Variables subset for Fault F-I detection selected by the PCMCI (partial correlation) method.

*Appendix A.6. Variables and Tags of the Real Industrial Case*

The variables of oil and gas metering process and its respective tags are listed in the table.

**Table A3.** Lisf of variables—Oil and gas fiscal metering station.

| Variable | Tag | Plant Section | Variable | Tag | Plant Section |
|---|---|---|---|---|---|
| Gas flow rate in processing 05 | FIP-05-D | D | Temperature of water output in cooler 02B | TI-02B-D | D |
| Level Tank 03 | LI-03-F | F | Temperature of water output in cooler 02A | TI-02A-D | D |
| Pump pressure 05 in oil transfer | PP-05-D | D | Flow rate in transfer oil 01B | FIT-01B-F | F |
| Temperature in treatment tank 01A | TI-01A-F | F | Flow rate in transfer oil 01A | FIT-01A-F | F |
| Flow rate for water treatmente 01A | FIT-01A-E | E | BSW in treatment tank outlet 02 | BSW-02O-D | D |
| Density in gas fiscal meter 01 | DR-01-A | A | BSW in treatment tank 01 | BSW-01-D | D |
| Specific mass in oil fiscal meter 01 | SM-01-C | C | Density in gas fiscal meter 03 | DR-03-B | B |
| BSW in treatment tank 02 | BSW-02-D | D | Temperature of oil output in cooler 01B | TI-01B-D | D |
| Flow of water treated | FW-E | E | Temperature of oil output in cooler 01A | TI-01A-D | D |
| Pump pressure 02B | PP-02B-D | D | Temperature of oil input in heat exchanger 02B | TI-02B-D | D |
| Pump pressure 02C | PP-02C-D | D | Oil flow rate 2 | FIO-2-E | E |
| Density in gas fiscal meter 03 | DR-03-A | A | Tank Pressure 01 | PI-01-F | F |
| Static pressure in gas fiscal meter 03 | PIT-03-B | B | Flow rate for water treatmente 01B | FIT-01B-E | E |
| Flow rage in gas fiscal meter 02A | FIT-02A-A | A | Temperature in treatment tank 01B | TI-01B-F | F |
| Pressure differential in gas fiscal meter 02A | PDIT-02A-A | A | Temperature of oil output in heat exchanger 01B | TI-02B-D | D |
| Pressure differential in gas fiscal meter 02B | PDIT-02B-A | A | Temperature of oil input in heat exchanger 01A | TI-01A-D | D |
| Static pressure in gas fiscal meter 02A | PIT-02A-A | A | Temperature of oil input in heat exchanger 01B | TI-01B-D | D |
| Static pressure in gas fiscal meter 02B | PIT-02B-A | A | Pump pressure 01B in oil transfer | PP-01B-D | D |
| Temperature in gas fiscal meter 02A | TIT-02A-A | A | Pump pressure 01A in oil transfer | PP-01A-D | D |
| Temperature in gas fiscal meter 02B | TIT-02B-A | A | Pressure differential in oil treatment tank 01 | PDIT-01-D | D |
| Gas flow rate | FIT-GC-G | G | Electric current in pump 07 | EC-07-D | D |
| Pump pressure in oil transfer | PP-0T-D | D | Electric current in pump 06 | EC-06-D | D |
| Controller output in wash tank 01 | CO-01-D | D | Flow injection in treament equipment 05 | FIP-05-D | D |
| Pump pressure 02A | PP-02A-D | D | Pump pressure for injection in Section D | PP-I-D | D |
| Oil flow rate 1A | FIO-1A-E | E | Pressure differential in importation gas | PDIT-IM-A | A |
| Oil flow rate 1 | FIO-1-E | E | Pressure in treatment tank 01B | PI-01B-F | F |
| Tank Pressure 02 | PI-02-F | F | Pressure in treatment tank 01A | PI-01A-F | F |
| Tank Pressure 03 | PI-03-F | F | Controller output in wash tank 01 | CO-01-D | D |
| Gas flow rate 1 | FIG-1-E | E | | | |

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
