# Peer review of "Variable Selection for Fault Detection Based on Causal Discovery Methods: Analysis of an Actual Industrial Case"

_processes, doi:10.3390/pr9030544_

Round 1
Reviewer 1 Report
The paper describes a methodology based an the randomness of the data. The results are certainly interesting. However more precise considerations could be made using more powerful mathematical algorithms, as can be read in the scientific literature. It is believed that the paper can be accepted after a minor revision concerning the description of the methodology used and the conclusions, with particular reference to future developments which, at the moment, are not clear.
Author Response
Thank you very much for your considerations! We agreed that that the methodology and conclusion sections (3.3 and 5 respectively) could be improved.
Therefore, we hope the following changes may clear out doubts on the methodology employed and lacks found in the paper.
- Specifically, a better description of methodology was met by adding the paragraphs defined by lines 202-218 and lines 226-231.
- And suggestions for future developments were made in the paragraph defined by lines 424-434.

Reviewer 2 Report
In this paper, a causality-based variable selection method is proposed and applied to fault detection. A variant of PC algorithm is used to extract causal relationship. The proposed method, along with many other variable selection methods, is applied to an industrial case (oil and gas fiscal metering station). Extensive results and analysis are provided to support the methodology.
Variable selection in high-dimensional data is very important in data analysis. This paper adds to the literature of causality-based variable selection with abundant empirical results.
One of the biggest contributions, it seems to me, is that a thorough analysis of causality-based variable selection methods and comparison with other methods in the application of industrial fault detection. Below are my thoughts on the side of statistical methodology.
A major flaw, in my opinion, is that the originality of the proposed robust PC algorithm is obscure. Following the description of Algorithm 1, there is not much for readers to learn how the new method compares with the original PC-algorithm. Some justification and analysis would be helpful for readers to understand the algorithm.
Also, the extensive data analysis is excellent; however, it may be somewhat overwhelming. Perhaps it would be better to highlight and summarize key improvements of the proposed method over existing ones.
(A typo in Algorithm 1: lenght --> length)
Author Response
Thank you very much for your considerations! We agreed that that the PC algorithms and final considerations sections (2.4 and 4.4 respectively) could be improved.
Therefore, we hope the following changes may clear out doubts on the and lacks found in the paper.
- Specifically, a better description of the main difference between the PC algorithm and the PC Stable algorithm was met by adding the paragraph defined by lines 152-157.
- A schematic example applying the PC algorithm and PC Stable algorithm was added in the new appendix A1. This section is addressed to show the main difference between the algorithms and the justification for employing the PC Stable algorithm in the work. Also, it would be helpful for readers to understand the algorithms.
-
Overall, it was considered 12 selection variable methods corresponding to 3 filter-based, 4 wrapped-based, 2 embedded-based, and 3 causality-based. Each selection variable method was applied in 5 fault detection scenarios, being 3 fault cases of the actual industrial case and 2 fault cases of TEP, using four different machine learning models.
In that way, would be needed 60 figures to show the model predictions (like Figure 2) and 60 figures to show the SPE index (like Figure 3). It led to represent and quantify the fault detection performance using the (FDR, FAR, and adjustment) metrics which were presented in Tables 7-11. Note that Figures 2-7 show the predictions and SPE index only related to the variable selection method that achieved the best performance in each fault case. We agree that the size and quantity of figures and tables may be somewhat overwhelming. However, we considered including the analysis of 3 different types of faults in the real industrial case in order to observe the fault detection engine behavior with failures with different signatures. Also, it seems important to us to include a benchmark analysis case addressed to allow the reproduction of the results since the real industrial datasets are untransferable according to company policies that gave them.
In order to help the readers we include the subsection 'Final considerations' (Section 4.4) to highlight and summarize key improvements in this work.

Reviewer 3 Report
The paper presents a comparison of variable selection approaches for fault detection based on causal discovery methods. The quality of the current version of the paper is generally about average.
Follows a list of comments to help the authors to improve the paper quality.
In Introduction, a paragraph at the end of this section, which describes the remainder of the paper, would have been helpful.
Figures 1 and A1 are missing.
Tables 5 and A3 should be resized to smaller ones.
I am not very happy with the amount and particularly the size of some tables and figures in the text. Perhaps some of them should be moved in the Appendices or even removed if they do not add much of useful information.
The section 4 Results finishes without providing neither a summary nor a discussion on the results obtained. I think that a discussion as a subsection (or even a new section) would be helpful.
Author Response
Thank you very much for your considerations! We agreed that the mentioned sections could be improved.
Therefore, we hope the following changes may clear out doubts on the and lacks found in the paper.
- Specifically, a paragraph at the end of the Introduction section describing the remainder of the paper was added in the paragraph defined by lines 88-94.
- Figures 1 and A1 correspond to the flow diagram of the real industrial case and the TEP process and were inserted in the manuscript that was submitted. However, in this first revision, we decided to save them in another format (without vectorization) to facilitate the compilation of the pdf file.
- Table 5 was resized to a smaller one (keeping the font size according to the Processes' author instructions).
- Table A3, (Appendix A6) was resized to a smaller one reducing the font size.
- Overall, it was considered 12 selection variable methods corresponding to 3 filter-based, 4 wrapped-based, 2 embedded-based, and 3 causality-based. Each selection variable method was applied in 5 fault detection scenarios, being 3 fault cases of the actual industrial case and 2 fault cases of TEP, using four different machine learning models. In that way, would be needed 60 figures to show the model predictions (like Figure 2) and 60 figures to show the SPE index (like Figure 3). It led to represent and quantify the fault detection performance using the (FDR, FAR, and adjustment) metrics which were presented in Tables 7-11. Note that Figures 2-7 show the predictions and SPE index only related to the variable selection method that achieved the best performance in each fault case. We agree that the size and quantity of figures and tables may be somewhat overwhelming. However, we considered including the analysis of 3 different types of faults in the real industrial case in order to observe the fault detection engine behavior with failures with different signatures. Also, it seems important to us to include a benchmark analysis case addressed to allow the reproduction of the results since the real industrial datasets are untransferable according to company policies that gave them.
- In order to help the readers we include the subsection 'Final considerations' (Section 4.4) to highlight and summarize key improvements and discussions in this work.

Reviewer 4 Report
The paper is well structured and written. I could not find any grave spelling or grammar errors. This greatly added to the reading pleasure.
The subject itself is - as it is mostly with case studies - is not very innovative and the finding that causality derived model parameters correlate better is not very suprising.
The readers intrest could be improved by discussing the application of the method applied to outer fields.
Author Response
Thank you very much for your considerations! We hope the following comments may clear out doubts and lacks found in the paper.
- We agreed that case studies research is restricted and the respective analysis, generally, is valid to particular cases. However, in this work, we studied fault detection scenarios corresponding to the real industrial cases providing the opportunity to work with issues rarely found in simulated or benchmark cases such as high dimensionality, real noised measures, and divergences between the fault events reported and the actual manifestation of the failure.
Selection variable approaches applied in experimental cases can be found in the literature. However, to the best of our knowledge, the strong point of these approaches is really tested in high-dimensional scenarios, which can only be achieved in industrial case scenarios (that are rarely found in the literature). -
In the present work, causality methods based used as variable selection approaches outperformed the other selection methods studied in this paper. However, this result was expected only in relation to the filter-based methods where the ranking of variables is established only by a score computation for each variable. Nevertheless, the variable subset found by these methods may not correspond to the subset that maximizes the classifier-regressor performance since variable relevance is computed one at a time.
In this way, the causality-based methods (that rigorously are filter-based methods) are not necessarily better than wrapped-based or embedded-based methods which are addressed to find the entire subset that maximizes the regressor performance.
- suggestions for future developments were made in the paragraph defined by lines 424-434. These future developments can include applications in outer fields. In our opinion, the subject of this work is directed to chemical industrial processes and given the number of results we consider it infeasible to include new cases in this work. Some external applications of causal methods can be found in several articles that were cited in the literature review and can eventually be studied in future works.

Round 2
Reviewer 3 Report
No further comments.